# Voxelated bioprinting of modular double-network bio-ink droplets

Jinchang Zhu ⬤[1], Yi He[2], Yong Wang[2] & Li-Heng Cai ⬤[1,3,4] ✉

Analogous of pixels to two-dimensional pictures, voxels—in the form of either small cubes or spheres—are the basic building blocks of three-dimensional objects. However, precise manipulation of viscoelastic bio-ink voxels in three-dimensional space represents a grand challenge in both soft matter science and biomanufacturing. Here, we present a voxelated bioprinting technology that enables the digital assembly of interpenetrating double-network hydrogel droplets made of polyacrylamide/alginate-based or hyaluronic acid/alginate-based polymers. The hydrogels are crosslinked via additive-free and biofriendly click reaction between a pair of stoichiometrically matched polymers carrying norbornene and tetrazine groups, respectively. We develop theoretical frameworks to describe the crosslinking kinetics and stiffness of the hydrogels, and construct a diagram-of-state to delineate their mechanical properties. Multi-channel print nozzles are developed to allow on-demand mixing of highly viscoelastic bio-inks without significantly impairing cell viability. Further, we showcase the distinctive capability of voxelated bioprinting by creating highly complex three-dimensional structures such as a hollow sphere composed of interconnected yet distinguishable hydrogel particles. Finally, we validate the cytocompatibility and in vivo stability of the printed double-network scaffolds through cell encapsulation and animal transplantation.

Analogous of pixels to two-dimensional (2D) pictures, voxels—in the form of either small cubes or spherical particles—are the basic units of three-dimensional (3D) objects[1]. A prominent example is Minecraft®, one of the world's best-selling video games. In the 3D-mosaic world of Minecraft®, everything—animals, houses, even the sun, and the moon—is made of small cubes. Players use these voxels with various functions and colors to construct their own artworks. The only limitation is the players' imagination. A similar approach, if realized in 3D cell assembly, would provide a standardized method for tissue engineering, transforming basic and translational biomedicine. In principle, one can use cell-encapsulated hydrogel droplets as voxels, assemble the voxels to create hierarchical and organized 3D structures, and exploit biophysical[2] and biochemical[3] cues to program the assemblies to highly functional 3D tissues. However, this approach requires precisely manipulating highly viscoelastic bio-ink voxels in 3D space, which represents a grand challenge in both soft matter science and biomanufacturing.

Recently, building on the advancement of embedded 3D printing[4–7], we proposed and showed the concept of a voxelated bioprinting technology that enables the digital assembly of spherical particles (DASP)[8,9]. DASP generates a highly viscoelastic aqueous droplet in an aqueous yield-stress fluid, deposits the droplet at a prescribed location, and assembles individual droplets into 3D structures by controlled polymer swelling (Fig. 1a). DASP is qualitatively different from existing embedded droplet printing technologies[10,11], which rely on the classic Rayleigh-Plateau instability to generate droplets. As a result, these techniques can only handle low-viscosity liquids and often involve biohazardous organic solvents; more importantly, they have no control over the absolute location of each droplet in 3D space. By

[1]Soft Biomatter Laboratory, Department of Materials Science and Engineering, University of Virginia, Charlottesville, VA 22904, USA. [2]Department of Surgery, University of Virginia, Charlottesville, VA 22903, USA. [3]Department of Chemical Engineering, University of Virginia, Charlottesville, VA 22904, USA. [4]Department of Biomedical Engineering, University of Virginia, Charlottesville, VA 22904, USA. ✉e-mail: liheng.cai@virginia.edu

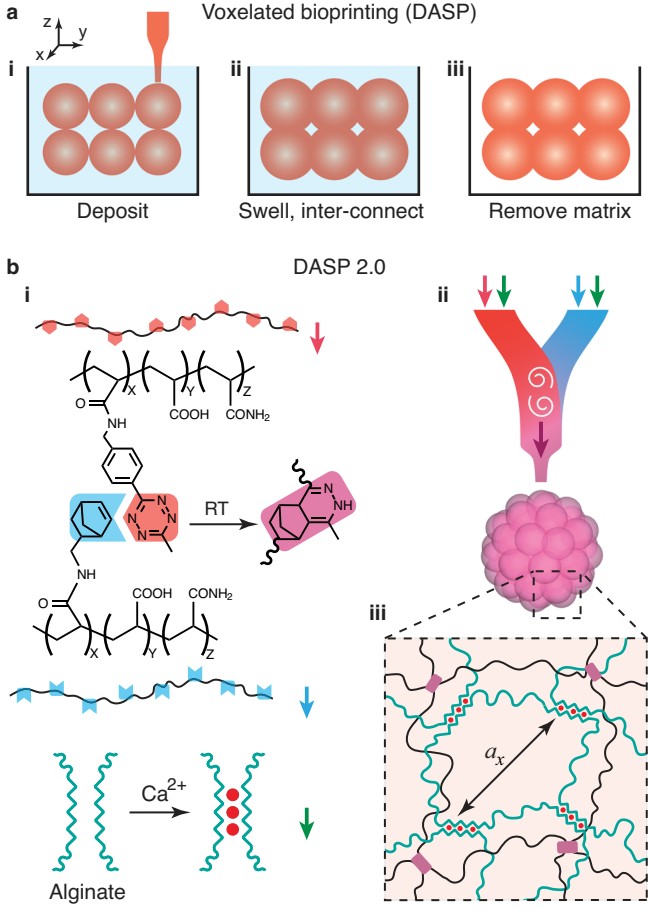

**Fig. 1 | Digital assembly of spherical particles (DASP) made of modular double-network bio-inks. a** DASP uses spherical bio-ink droplets, not conventional one-dimensional (1D) bio-ink filaments, as the building blocks to create three-dimensional (3D) objects. In DASP printing, (i) droplets are deposited at precisely controlled locations and with prescribed volume in a sacrificial supporting matrix made of an aqueous yield-stress fluid. (ii) Under controlled swelling, the deposited droplets partially coalesce with their neighbors. (iii) Following a complete cross-linking of droplets and the removal of the sacrificial matrix, the printed particles form a free-standing 3D construct made of interconnected yet distinguishable hydrogel particles. **b** DASP 2.0 enables printing modular double-network (DN) bio-inks. The ink consists of alginate and poly(acrylamide) (PAM), which is functionalized with either norbornene (NB) or tetrazine (TZ) groups. (i) In an aqueous solution, the NB-PAM (blue) and TZ-PAM (red) form a network via click reaction between NB and TZ at RT without any external trigger, while alginate forms a network via ionic crosslinking. (ii) A custom-designed multi-channel print nozzle is developed to enable printing DN bio-inks. The nozzle mixes a pair of bio-inks, TZ-PAM (red arrow)/alginate (green arrow) + NB-PAM (blue arrow)/alginate (green arrow), homogeneously without damaging cells. (iii) In a DN hydrogel, the ionically crosslinked alginate network interpenetrates with the PAM network.

contrast, DASP exploits nonlinear fluid dynamics to precisely manipulate bio-ink droplets in 3D space. The mechanism of DASP printing is somewhat reminiscent of the "tablecloth trick" that involves pulling a loaded tablecloth away from a table but leaving the plates behind. This trick relies on inertia. The key difference is that, in DASP, the "inertia" is not from the droplet but from the confinement force from the supporting matrix, so that the embedded droplet can be deposited at a prescribed location. Compared to the printing of spheroids[12–14], in which solid-like cell aggregates are stacked with great care and subsequently fused through slow cell migration, in DASP the droplets are joined by relatively fast and more controllable polymer swelling. Thus, DASP provides a robust approach for the on-demand assembly of cell-encapsulated hydrogel particles in a cytocompatible environment.

Using DASP, we created multiscale porous scaffolds formed by interconnected yet distinguishable alginate hydrogel particles. In our proof-of-concept research, we encapsulated human islets in individual particles and demonstrated that the scaffolds allow for responsive insulin release[8], highlighting the potential of DASP in biomedical applications. However, our prototype voxelated bioprinting technology can print pure alginate hydrogels only, which are brittle and have limited tunability in mechanical properties. These limit the applications of DASP not only in translational biomedicine, such as printing mechanically robust scaffolds as long-term transplants, but also in basic biomedical research, which requires modular biomaterials for controlling the microenvironment of voxels.

Here, we develop a substantially improved voxelated bioprinting technology (DASP 2.0) that enables the digital assembly of interpenetrating alginate and polyacrylamide (PAM) double-network (DN) bio-ink droplets (Fig. 1b). We design and synthesize a pair of PAM polymers functionalized with tetrazine (TZ) and norbornene (NB) groups, respectively. Upon mixing, this pair of TZ-PAM and NB-PAM (A + B) polymers form a network via additive-free click reaction between TZ and NB groups (Fig. 1b, i). For the single-network PAM hydrogels, we systematically investigate the dependence of cross-linking kinetics and stiffness on NB/TZ ratio and polymer concentration. The experimentally observed behavior can be well described by kinetic and scaling theories. A diagram-of-state is constructed to outline the stiffness and extensibility of DN PAM/alginate ((A + B)/C) hydrogels with various formulations. We engineer multi-channel print nozzles for on-demand mixing of highly viscoelastic bio-inks without significantly impairing cell viability (Fig. 1b, ii). Further, we demonstrate the capability of voxelated bioprinting by creating highly complex 3D structures such as a hollow sphere composed of interconnected yet distinguishable hydrogel particles. The voxel-printed DN scaffolds can repeatedly sustain a large extent of compression up to 60% without noticeable damage. Moreover, we transplant the DN scaffolds into abdominal cavity of immunocompetent mice to show that the scaffolds can be retrieved after 4 months with negligible impaired structural integrity and mechanical properties. Finally, we demonstrate the universality of using (A + B)/C type DN hydrogels as inks for DASP by replacing the polymer backbone of the A + B network from PAM to hyaluronic acid (HA) without compromising mechanical robustness. The developed cytocompatible DN biomaterials (Fig. 1b, iii) and printing systems pave the way for voxelated bioprinting highly complex yet organized tissue constructs for basic and translational biomedicine.

## Results

### Crosslinking kinetics of single-network PAM hydrogels

To be suitable for DASP printing, the bio-ink should possess a gelation time of sufficient duration. This allows the printed droplets to undergo partial swelling, enabling them to partially coalesce with adjacent droplets before complete crosslinking occurs (Fig. 1a). However, the crosslinking time should not be too long, otherwise the bio-ink may diffuse through the porous supporting matrix, leading to uncontrollable droplet shape[8]. Thus, it is essential to quantify the crosslinking kinetics of single-network PAM hydrogels.

Unlike typical hydrogels formed by crosslinking precursor polymer chains using small molecules as crosslinking agents, our single-network PAM gels are formed by crosslinking a pair of linear PAM polymers. The two polymers are of the same molecular weight (MW) but are functionalized by tetrazine (TZ) (Supplementary Fig. 3) and norbornene (NB) groups, respectively. Upon reaction, the TZ and NB groups form a covalent bond via additive-free click reaction at room temperature (RT) (Fig. 1b, i). Therefore, the crosslinking kinetics of our single-network PAM hydrogels is expected to depend on both the concentration, $c$, of the polymers and the grafting ratio, $f_x$, of the functional groups, which is defined as the molar ratio between the

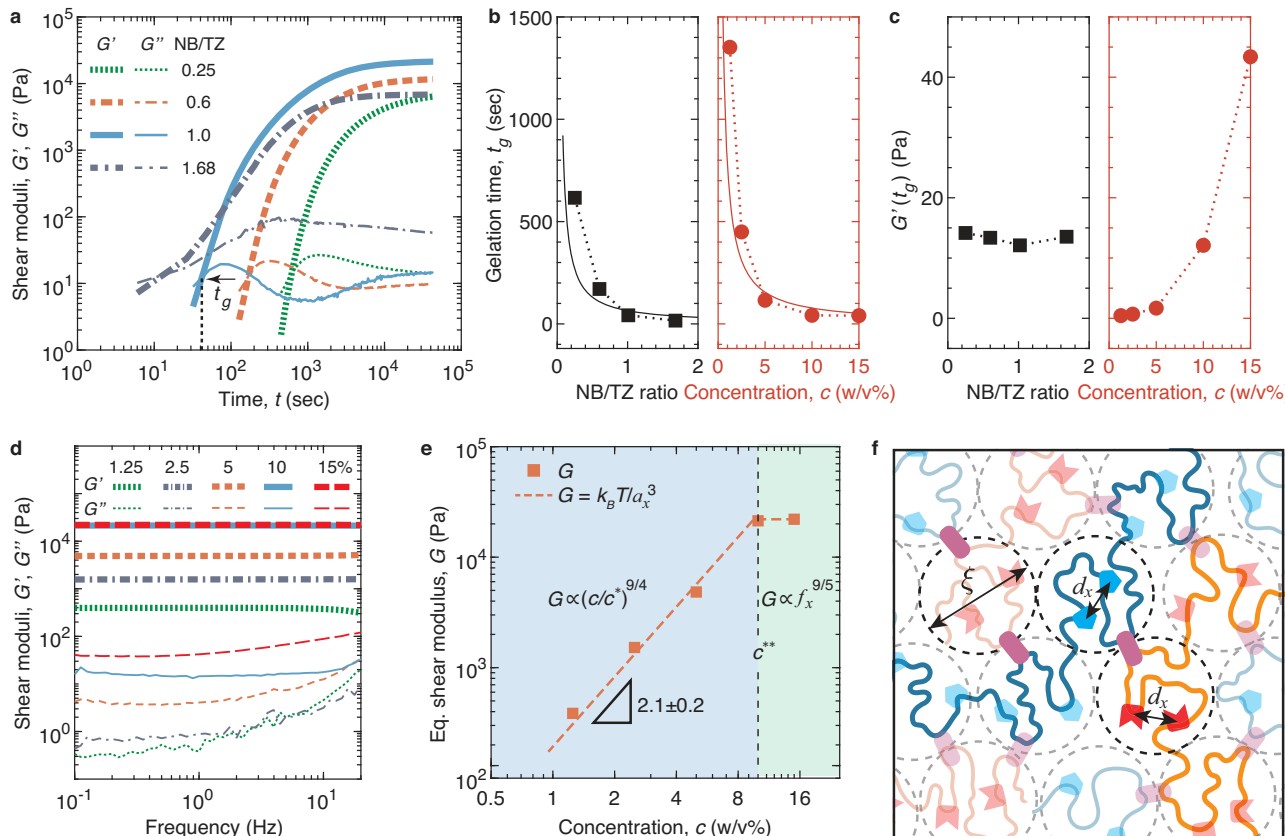

**Fig. 2 | Crosslinking kinetics and stiffness of single-network PAM hydrogels crosslinked through biofriendly click chemistry. a** Real-time characterization of storage ($G'$, thick lines) and loss ($G''$, thin lines) moduli for mixtures of NB-PAM and TZ-PAM at various molar ratios between NB and TZ groups. The concentrations of TZ-PAM and NB-PAM solutions are both fixed at 10% (w/v) before mixing. For TZ-PAM, the grafting ratio of TZ is fixed at 2.53%, defined as the molar ratio between the grafted tetrazine and the total number of chemical monomers on the PAM backbone. This value corresponds to on average 52 TZ groups per polymer. In contrast, for NB-PAM, the grafting ratio of NB groups varies from 0.63 to 4.25%. The gelation time, $t_g$, is defined as the point above which $G'$ surpasses $G''$. All measurements are conducted at RT, using a fixed strain 0.5% and an oscillatory shear frequency 1 Hz. **b** The dependencies of gelation time, $t_g$, on the NB/TZ ratio at fixed polymer concentration of 10% (w/v) and on the polymer concentration, $c$, at fixed NB/TZ ratio of 1. **c** The dependencies of storage modulus, $G'$, at gelation time, $t_g$, on the NB/TZ ratio at fixed polymer concentration of 10% (w/v) and on the polymer concentration, $c$, at fixed NB/TZ ratio of 1. **d** The dependencies of storage ($G'$, thick lines) and loss ($G''$, thin lines) moduli of a completely crosslinked single-network PAM hydrogels on the oscillatory shear frequency. In all hydrogels, both the TZ-PAM and NB-PAM polymers have roughly the same grafting ratio of 2.5%, and they are mixed at the same concentrations ranging from 1.25 to 15% (w/v). **e** The dependencies of the experimentally measured (solid squares) and the theoretically predicted (dash line) shear modulus $G$ for PAM hydrogels on the concentration, $c$, of PAM polymers. $c^*$, overlap concentration of PAM solutions; $c^{**}$, the crossover contrast at which the correlation length, $\xi$, is about the average distance between two neighboring functional groups on the same the polymer chain, $d_x$. For $c > c^{**}$, $\xi < d_x$. **f** Schematic of our theory explaining the relation between the concentration of PAM polymers and the network shear modulus. In a crosslinked network, the mesh size $a_x$ exhibits two distinct regimes: (1) for $c < c^{**}$, $a_x \approx \xi$; (2) for $c > c^{**}$, $a_x$ becomes saturated and $a_x \approx d_x$. The network modulus $G \approx k_B T / a_x^3$, as indicated by the dashed line in (**e**).

grafted functional groups and the total number of chemical monomers on the PAM backbone.

To investigate the crosslinking kinetics of PAM hydrogels, we maintain the concentrations of TZ-PAM and NB-PAM solutions at 10% (w/v) before mixing while varying the molar ratio between NB and TZ groups. To do so, we keep the grafting ratio of TZ-PAM, $f_{x,TZ}$, at 2.53% (Supplementary Fig. 4); on average, this value corresponds to approximately 50 TZ groups per polymer. We vary the grafting ratio of NB groups, $f_{x,NB}$, from 0.63% to 4.25% (Supplementary Fig. 5). These formulations cover NB/TZ ratios spanning approximately one order of magnitude, ranging from 0.25 to 1.68. For each mixture with equal amount of TZ-PAM and NB-PAM polymers, we employ a stress-controlled rheometer to continuously monitor its viscoelasticity in real time for 12 h (see Methods). We determine the gelation time, $t_g$, above which the storage modulus, $G'$, exceeds the loss modulus, $G''$, as denoted by the arrow in Fig. 2a. As the NB/TZ ratio increases from 0.25 to 1.68, $t_g$ decreases by nearly 40 times from 616 s to 16 s (solid squares in Fig. 2b).

The decrease in the gelation time with the increase of NB/TZ ratio can be explained by a simple kinetic theory. The rate of forming crosslinks, $dC_x/dt$, is proportional to the probability of a NB group to meet with a TZ group, which is the product of the concentrations, $(C_{NB,0} - C_x)$ and $(C_{TZ,0} - C_x)$, of the two in the mixture:

$$\frac{dC_x}{dt} = k(C_{NB,0} - C_x)(C_{TZ,0} - C_x) \tag{1}$$

Here, $k$ is a constant determined by the reaction rate between a NB and a TZ, and $C_{NB,0}$ and $C_{TZ,0}$, respectively, are the concentrations of unreacted NB and TZ groups at reaction time $t = 0$. Integrating Eq. (1), one obtains the relation between the concentration of crosslinks and reaction time:

$$t = \frac{1}{k} \begin{cases} \frac{1}{(C_{NB,0} - C_{TZ,0})} \ln\left[\frac{(C_{NB,0} - C_x)C_{TZ,0}}{(C_{TZ,0} - C_x)C_{NB,0}}\right], & C_{NB,0} \neq C_{TZ,0} \\ \frac{C_x}{C_0(C_0 - C_x)}, & C_{NB,0} = C_{TZ,0} = C_0 \end{cases} \tag{2}$$

At the gelation time $t_g$, each polymer has on average one cross-link, forming a giant molecule that percolates the whole volume of the solution[15]. As each polymer carries about 50 functional groups, the concentration of crosslinks relative to the initial concentration of the functional groups, $\beta \equiv C_{x,g}/C_{TZ,0} \approx 1/50$. Substituting this relation to Eq. (2), one obtains the dependence of the gelation time on the ratio between NB and TZ groups, $C_{NB,0}/C_{TZ,0}$:

$$t_g = \frac{1}{k} \begin{cases} \frac{1/C_{TZ,0}}{(C_{NB,0}/C_{TZ,0}-1)} ln\left[\frac{C_{NB,0}/C_{TZ,0}-\beta}{(1-\beta)C_{NB,0}/C_{TZ,0}}\right], & C_{NB,0} \neq C_{TZ,0} \\ \frac{\beta}{C_0(1-\beta)}, & C_{NB,0} = C_{TZ,0} = C_0 \end{cases} \quad (3)$$

Equation (3) predicts that, for $C_{NB,0} \neq C_{TZ,0}$, $t_g$ decreases rapidly with the increase of $C_{NB,0}/C_{TZ,0}$. This theory explains well the experimentally observed gelation time with the fitting parameter $k \approx 1.8 \times 10^{-2}$ L mol$^{-1}$ s$^{-1}$, as shown by the solid line on the left panel of Fig. 2b.

For $C_{NB,0} = C_{TZ,0} = C_0$, Eq. (3) predicts that the gelation time is inversely proportional to the initial concentration of functional groups. To test this prediction, we synthesize TZ-PAM and NB-PAM polymers with grafting ratios of 2.53% and 2.57%, respectively; this ensures that $C_{NB,0}/C_{TZ,0} \approx 1$. We prepare a series of solution mixtures consisting of equal amount of TZ-PAM and NB-PAM polymers but increase the polymer concentration $c$ from 1.25 to 15% (w/v). The experimentally measured gelation time decreases from 1352 to 40 s (solid circles in Fig. 2b). This behavior can be well explained by our theory, as shown by the solid line on the right panel of Fig. 2b. Moreover, the fitting parameter, $k \approx 1.5 \times 10^{-2}$ L mol$^{-1}$ s$^{-1}$, agrees reasonably well with that for $C_{NB,0} \neq C_{TZ,0}$.

The gelation theory is based on the physical picture that at the gelation point there is on average one crosslink per polymer. This implies two consequences for the hydrogel stiffness at the gelation point: (i) The stiffness should be constant if the polymer concentration is the same; (ii) the stiffness increases with the polymer concentration. Indeed, these predictions are verified by our experiments. For the hydrogel formulations with the same polymer concentration but various NB/TZ ratios, the shear storage modulus at the gelation time, $G'(t_g)$, is nearly a constant of 13 Pa (left panel in Fig. 2c). In contrast, for the hydrogel formulations of the same NB/TZ ratio but various polymer concentrations, $G'(t_g)$ increases as the polymer concentration increases (right panel in Fig. 2c). Taken together, our results provide the scientific foundation for prescribed gelation time of the single-network PAM hydrogels in a wide range of concentrations.

## Stiffness of single-network PAM hydrogels

To explore the stiffness of single-network PAM hydrogels, we fix TZ-PAM and NB-PAM polymers at a stoichiometrically matched grafting ratio of 2.5% and mix them at equal concentrations from 1.25% to 15% (w/v). We wait for sufficiently long enough time to allow the hydrogel to be completely crosslinked, as exemplified by the saturated $G'$ in Fig. 2a. For all hydrogel formulations, $G'$ is nearly independent of the frequency within the range of 0.1–20 Hz (thick lines, Fig. 2d). Moreover, $G'$ is more than 10 times greater than the loss modulus $G''$ (thin lines, Fig. 2d). These results indicate that the PAM hydrogels are elastic networks; therefore, one can take $G'$ at the lowest frequency as the equilibrium network shear modulus $G$.

Interestingly, the dependence of hydrogel stiffness on concentration exhibits two distinct regimes. As the polymer concentration increases from 1.25 to 10% (w/v), $G$ increases rapidly from 385 Pa to $2.1 \times 10^4$ Pa by a power law with an exponent of $2.1 \pm 0.2$ (dashed orange line, Fig. 2e). Further increasing the polymer concentration results in a negligible increase in stiffness (Fig. 2d, e). This two-regime behavior contradicts conventional understanding that the hydrogel stiffness increases with polymer and crosslinker concentrations.

We develop a scaling theory to describe the dependence of hydrogel stiffness on the functionality $f_x$ and concentration $c$ of PAM polymers. To form a crosslink, a TZ must meet a NB. Thus, the crosslink concentration is determined by the average distance between a TZ and a NB group. This distance also equals the correlation length, $\xi$, which is defined as the nearest distance of a monomer from another monomer on the neighboring polymer chains, as indicated by the dashed circles in Fig. 2f. Hence, the size of a network strand, $a_x$, is approximately the correlation length: $a_x \approx \xi$. The shear modulus of the network is about $k_B T$ per volume pervaded by a network strand:

$$G \approx k_B T/a_x^3 \approx k_B T/\xi^3 \quad (4)$$

Because water is a good solvent for PAM, $\xi$ decreases by $\xi \approx R_F(c/c^*)^{-3/4}$, where $c^*$ is the polymer overlap concentration, and $R_F \approx bN^{3/5}$ is the end-to-end distance of a single PAM chain in a dilute solution with $N$ being the number of Kuhn monomers per polymer[15]. This gives the dependence of the network shear modulus on polymer concentration:

$$G \approx \frac{k_B T}{R_F^3}\left(\frac{c}{c^*}\right)^{9/4} \quad (5)$$

However, this power law behavior will stop at concentrations greater than $c^{**}$, at which the correlation length $\xi(c^{**})$ is about the average distance, $d_x$, between two neighboring functional groups (TZ or NB) on the same polymer as marked in Fig. 2f. In this case, all functional groups are reacted, and the network shear modulus saturates at:

$$G_{max} \approx \frac{k_B T}{d_x^3} \quad (6)$$

In a good solvent,

$$d_x \approx b\left(f_x^{-1}/n_K\right)^{3/5} \quad (7)$$

Here, $b = 1.6$ nm is the Kuhn length of PAM, $n_K = \frac{1}{2}C_\infty/\cos^2\left(\frac{\theta}{2}\right) \approx 6.5$ is the number of chemical monomers per Kuhn monomer with the Flory's characteristic ratio $C_\infty = 8.5$ and the main C-C chain bond angle $\theta = 68°$[16], and $f_x^{-1} = 40$ is the average number of chemical repeating units within the polymer section of size $d_x \approx 4.8$ nm. Substituting Eq. (7) into (6), one obtains

$$G_{max} \approx \frac{k_B T}{\left[b\left(f_x^{-1}/n_K\right)^{3/5}\right]^3} \approx \frac{k_B T}{b^3}\left(n_K f_x\right)^{9/5} \propto f_x^{9/5} \quad (8)$$

Thus, for a fixed $f_x$, we expect two regimes for stiffness:

$$G \approx \begin{cases} \frac{k_B T}{R_F^3}\left(\frac{c}{c^*}\right)^{9/4} \propto (c/c^*)^{9/4}, & c^* < c < c^{**} \\ \frac{k_B T}{b^3}\left(n_K f_x\right)^{9/5} \propto f_x^{9/5}, & c > c^{**} \end{cases} \quad (9)$$

At relatively low concentrations, $c^* < c < c^{**}$, the network stiffness increases with the polymer concentration by a power of 9/4. In contrast, at high concentration, $c > c^{**}$, the network stiffness saturates and becomes a constant. The theoretical predictions [see Eq. (9)] explain the experiments remarkably well (Fig. 2e). Note that scaling theory does not include prefactors, which typically have values on the order of unit. For instance, our theory predicts that the maximum hydrogel stiffness is $G_{max} \approx 37$ kPa [Eq. (8)]. Comparing the prediction against the experimentally measured value 22 kPa, one obtains a prefactor of 0.6. Nevertheless, the combination of experiments and theory shows that the TZ/NB-PAM single-network hydrogels allow for prescribed

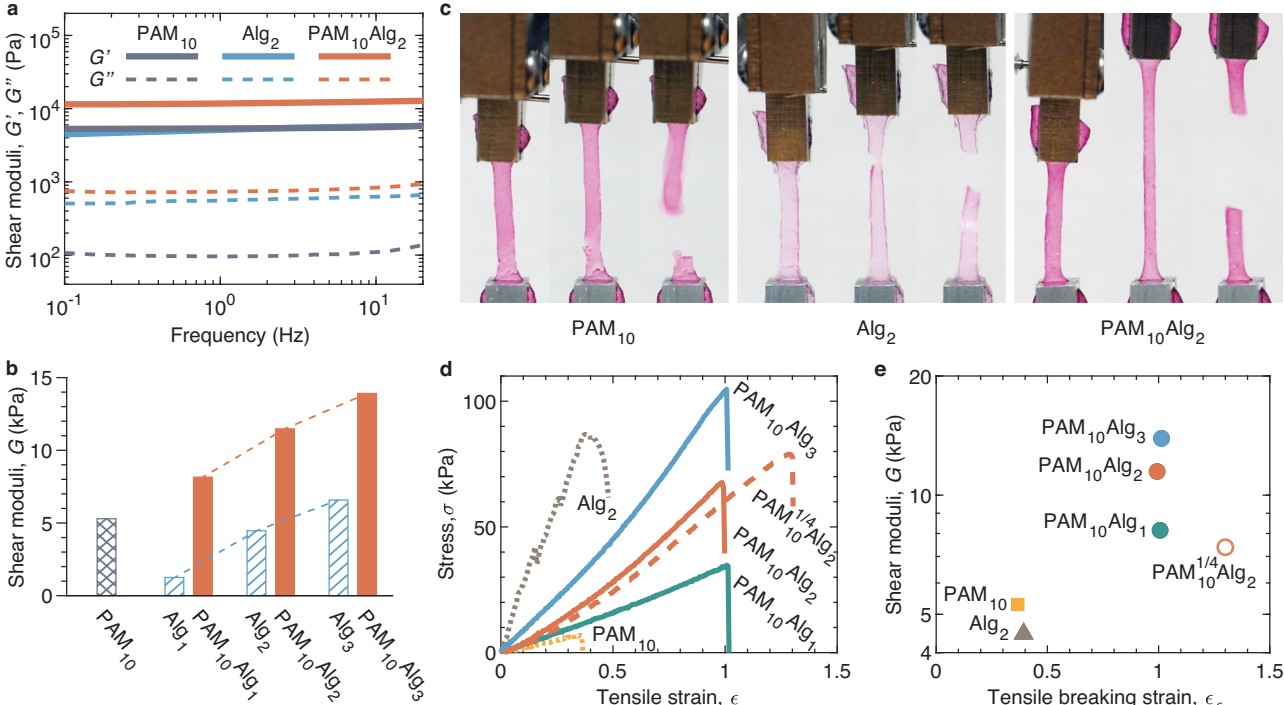

**Fig. 3 | Stiffness and extensibility of DN polyacrylamide/alginate (PAM/Alg) hydrogels.** The grafting ratio of TZ-PAM and NB-PAM is fixed at 2.5% unless otherwise specified. **a** Dependencies of storage ($G'$, solid lines) and loss ($G''$, dashed lines) moduli of completely crosslinked single-network PAM, single-network alginate (Alg), and double-network PAM/alginate hydrogels on oscillatory shear frequency. The composition of a crosslinked hydrogel is denoted as $PAM_xAlg_y$, where $x$ is the concentration in (w/v)% for PAM consisting of equal amount of TZ-PAM and NB-PAM, and $y$ is the concentration of alginate. **b** Equilibrium shear moduli of single-network PAM, single-network alginate, double-network PAM/alginate hydrogels. **c** Photos of $PAM_{10}$, $Alg_2$, and $PAM_{10}Alg_2$ hydrogels under uniaxial tensile tests at a fixed strain rate of 0.02/s. For each formulation, the left, middle, and right panels are, respectively, captured at the beginning of the tensile test, right before breaking, and right after breaking. Samples are 13 mm in length and 2 mm in width of the center part. **d** Stress–strain behavior of hydrogels. Dashed red line: a $PAM_{10}Alg_2$ double-network hydrogel made from TZ-PAM with a grafting ratio of 2.53% and NB-PAM with a grafting ratio of 0.63%, which is denoted as $PAM_{10}^{1/4}Alg_2$. **e** A two-parameter ($G$, $\epsilon_f$) diagram-of-state that outlines the mechanical properties of all hydrogels formulations in (**d**). Empty circle corresponds to the hydrogel $PAM_{10}Alg_2$ with mismatched TZ and NB grafting ratios in (**d**), which is denoted as $PAM_{10}^{1/4}Alg_2$.

stiffness from ~400 Pa to 22 kPa, sufficient for cell encapsulation and most biomedical applications[16–18].

## Stiffness and extensibility of DN hydrogels

To prepare a DN hydrogel, we use TZ-PAM and NB-PAM, respectively, to prepare two solutions, dissolve the same amount of alginate (Alg) in each of the two solutions, mix the pair of the solutions to initiate the crosslinking of PAM network, and then transfer the partially crosslinked hydrogel to a bath containing 20 mM $Ca^{2+}$ to incubate for 10 min at RT to completely crosslink both the alginate and PAM networks. We denote a DN hydrogel formulation as $PAM_xAlg_y$, where $x$ is the concentration in (w/v)% for TZ-PAM or NB-PAM before mixing, and $y$ is the concentration of alginate. In all formulations, we use TZ-PAM and NB-PAM with stoichiometrically matched grafting ratio of 2.5% at a concentration of 10% (w/v); this formulation results in a single-network PAM hydrogel $PAM_{10}$ of shear modulus approximately 20 kPa (Fig. 2d), comparable to that of most organs in abdominal space[19].

Because the alginate network contains carboxylic acid groups and the PAM network contains carboxylic acid and amide groups, and that both the functional groups are sensitive to ions and pH, we equilibrate and completely crosslink hydrogel in cell culture media (DMEM) with 2 mM $CaCl_2$ for 24 h before characterizing its mechanical properties. Indeed, all the hydrogel samples swell in DMEM, as exemplified by a swelling ratio of 7% in size for $PAM_{10}Alg_2$ (Supplementary Fig. 8). The shear modulus of $PAM_{10}$ hydrogel decreases dramatically from 21 kPa in water to 5.3 kPa in cell culture media. Yet, the hydrogel remains an elastic solid, as evidenced by the nearly frequency-independent $G'$ as well as small loss factors, $\tan\delta \equiv G''/G' \approx 0.02$, within the frequency

ranging from 0.1 to 20 Hz (solid gray line, Fig. 3a). Similarly, frequency-independent dynamic moduli are observed for 2% (w/v) alginate hydrogel, $Alg_2$, and DN hydrogels (blue and red lines, Fig. 3a). These results show that all hydrogels are non-dissipative, elastic solids.

The shear modulus of a DN hydrogel is approximately the sum of the moduli of its constituent singlet-network hydrogels. For instance, $Alg_2$ and $PAM_{10}$ exhibit shear moduli of 4.5 kPa and 5.3 kPa, respectively, resulting in a combined modulus of 9.8 kPa, which closely matches the value of 11 kPa for $PAM_{10}Alg_2$. This observation holds true for other DN hydrogel formulations as well. When the concentration of alginate increases from 1% to 3% (w/v), the stiffness of single-network alginate hydrogels increases linearly from 1.2 kPa to 6.6 kPa (dashed bars, Fig. 3b). The corresponding DN hydrogels show a similar increase in stiffness from 8.2 kPa to 14 kPa with a nearly the same magnitude (solid bars, Fig. 3b).

Next, we perform uniaxial tensile tests to quantify the extensibility of DN hydrogels. To do so, we take a fully crosslinked hydrogel film measuring approximately 1 mm in thickness and cut it into a dog-bone shape sample. We then subject the sample to a fixed strain rate of 0.02/s while capturing the process using a camera, as visualized by the photos in Fig. 3c and by Supplementary Movie 1. For all hydrogels, we find that the stress increases almost linearly with tensile strain until the hydrogels reach their breaking point. However, we note that the single-network $Alg_2$ hydrogel is often too brittle for reliable measurements. These findings further confirm that the hydrogels are nearly pure elastic networks with little energy dissipation.

The DN $PAM_{10}Alg_2$ hydrogel exhibits significantly improved extensibility compared to the single-network $PAM_{10}$ hydrogel. While

$PAM_{10}$ has a tensile breaking strain $\epsilon_f$ of 0.36, the $PAM_{10}Alg_2$ hydrogel demonstrates approximately three times greater extensibility with $\epsilon_f$ of 1.0, as shown in Fig. 3c, d and by Supplementary Movie 1. Remarkably, although increasing the alginate concentration stiffens the corresponding DN hydrogels, $\epsilon_f$ remains nearly the same at 1 (solid lines in Fig. 3d and filled circles in Fig. 3e).

We attribute the observed enhancement in network extensibility to the characteristics of DN hydrogels. Upon deformation, the weak alginate network fractures first; this process effectively prevents localized and amplified stress along PAM network strands, thereby improving network extensibility. However, the network extensibility cannot exceed that afforded by the network strand of the PAM network. In theory, the maximum elongation at break, $\lambda_{max}^T$, for a network strand equals the ratio between its initial size, $a_x$, and contour length, $L_{max}$:

$$\lambda_{max}^T = \frac{L_{max}}{a_x} \tag{10}$$

However, typical single-network hydrogels, which often have a wide distribution in network strand size, cannot reach this maximum extensibility. When subjected to deformation, the relatively short network strands would break first, resulting in extensibility lower than the prediction based on average network size $a_x$. By contrast, in a DN hydrogel, the weak network can undergo fracture to prevent localized, amplified stress near network defects or along network strands, and therefore, avoid premature failure of the strong network. Consequently, a DN hydrogel may reach the theoretical extensibility $\lambda_{max}^T$[20,21].

We develop a scaling theory to describe the dependence of network theoretical extensibility $\lambda_{max}^T$ on network stiffness $G$. In a DN PAM/alginate hydrogel, the PAM network has on average $N_x$ Kuhn monomers per network strand. In a good solvent, the end-to-end distance, or the size of the network strand, is a self-avoiding random walk of Kuhn monomers with size of $b$:

$$a_x \approx bN_x^{3/5} \tag{11}$$

Recall Eq. (10), the theoretical extensibility of the network is

$$\lambda_{max}^T \approx N_x^{2/5} \tag{12}$$

For an unentangled network, its shear modulus $G$ is about $k_BT$ per volume pervaded by the network strand,

$$G \approx \frac{k_BT}{a_x^3} \approx \frac{k_BT}{b^3 N_x^{9/5}} \tag{13}$$

Thus, one can correlate $\lambda_{max}^T$ to the network stiffness $G$,

$$\lambda_{max}^T \approx (k_BT/G)^{2/9} b^{-2/3} \propto G^{-2/9} \tag{14}$$

Equation (14) predicts that extensibility of the DN hydrogel increases with the decrease of PAM hydrogel stiffness.

To test this prediction, we synthesize another DN hydrogel, $PAM_{10}^{1/4}Alg_2$, in which the TZ-PAM has a grafting ratio of 2.53%, while the NB-PAM has a mismatched grafting ratio of 0.63%. This formulation results in a DN hydrogel with lower network shear modulus, $G \approx 7.4$ kPa (Supplementary Fig. 9) and higher maximum extensibility, $\lambda_{max,1/4} \approx 2.3$ (dashed line in Fig. 3d and empty circle in Fig. 3e). Considering that the ionically crosslinked alginate network has negligible contribution to $\lambda_{max}^T$ and the equilibrium shear modulus $G_{1/4}$, $G_{1/1}$, and $G_{alg}$ are respectively 7.4, 11.5, and 4.5 kPa (Fig. 3b, e), the ratio of the extensibility of the softer DN hydrogel with NB/TZ ratio 1/4 to that of the stiffer one with NB/TZ ratio 1/1 is predicted to be $\lambda_{max,1/4}^T/\lambda_{max,1/1}^T \approx [(G_{1/4} - G_{alg})/(G_{1/1} - G_{alg})]^{-2/9} \approx 1.2$. Remarkably,

this value agrees well with the experimentally measured ratio 1.2. These results collectively show that the extensibility of the DN hydrogel is determined by the PAM network.

Notably, compared to the classical DN PAM/alginate hydrogel of the same composition, which exhibits a remarkable extensibility with $\epsilon_f \approx 20$[22], our DN hydrogel is much less stretchable. This significant difference in absolute extensibility is likely due to the difference in network topology. In the classical DN hydrogel, the PAM network is formed through in situ free radical polymerization of acrylamide monomers in an alginate solution. This polymerization is an uncontrolled reaction, resulting in a wide distribution in network strand size and leading to relatively large extensibility of a single-network PAM hydrogel with $\epsilon_f \approx 6.5$. In contrast, in our DN hydrogels, the PAM network is formed through the click reaction between TZ-PAM and NB-PAM precursor polymers, which have a relatively low and fixed molecular weight. Furthermore, the reactive groups further segment the linear PAM polymers into small sections, thereby further reducing the network's extensibility.

## DASP printing of DN hydrogels

DASP printing requires highly viscous and shear-thinning bio-inks, such that during printing, an extruded droplet can grow uniformly in the yield-stress fluid supporting matrix[9]. We conduct shear tests to measure the viscosity of the DN $PAM_{10}Alg_2$ bio-ink and its constituent $PAM_{10}$ and $Alg_2$ solutions. The $PAM_{10}$ solution is nearly a Newtonian fluid with a low viscosity of 0.2 Pa·s, which is nearly independent of shear rate ranging from 0.1 to 1000 s$^{-1}$, as shown by the squares in Fig. 4a. In contrast, the $Alg_2$ solution is a shear-thinning fluid, with the apparent viscosity decreasing with the shear rate by a power of −0.63, as shown by the triangles in Fig. 4a. However, the magnitude of the viscosity, 2.6 Pa·s, remains much lower than ~20 Pa·s required for DASP printing. Consequently, both constituent bio-inks are unsuitable for DASP printing. Remarkably, the viscosity of the DN bio-ink significantly increases to 17.1 Pa·s while maintaining shear-thinning behavior, as shown by the solid circles in Fig. 4a. These properties make the DN bio-ink $PAM_{10}Alg_2$ suitable for DASP printing[9]. Moreover, the crosslinked DN hydrogel has a significantly improved extensibility compared to its single-network constituents and a stiffness comparable to that of the organs in the abnormal cavity. Thus, we use $PAM_{10}Alg_2$ as the DN bio-ink for the subsequent DASP printing.

For DASP printing, the bio-inks are required to be extruded through the nozzle in a liquid state and rapidly crosslink once being deposited. Due to the rapid crosslinking of DN bio-inks upon mixing (Fig. 2a), premixing and loading the DN bio-inks into a single syringe, as performed in DASP 1.0, would clog the print nozzle and syringe. To avoid the clogging, we separately load the TZ-PAM and the NB-PAM in two syringes and then mix them on-demand during the printing. To do so, we engineer a dual-inlet print nozzle with an inner static mixer chamber (Supplementary Fig. 10). The design of the mixer chamber utilizes a structure called Quadro™ Square, as depicted by a computer-rendered 3D model in Fig. 4b, i. We use stereolithography printing to fabricate this nozzle, which has an outer diameter of 2.7 mm. The inner channel of the nozzle has a narrowest dimension of 425 μm. These dimensions are much larger than the size of typical cells, ~30 μm[23], and even some large cell aggregates such as islets, ~200 μm[24]. This design avoids mechanical shear-induced damage to the cells during extrusion.

To evaluate the mixing capability of the dual-inlet print nozzle, we use a camera to monitor the mixing process in real-time. To visualize the mixing, we incorporate red and green mica microparticles with the diameter of 50 μm into the respective bio-inks. The pair of bio-inks is extruded through the two inlets of the nozzle at a flow rate of 0.15 μL/s, which is the same as that used in DASP printing. Initially, we turn on only the red channel, allowing the mixer chamber to be filled with red particles (left, Fig. 4b, ii). Subsequently, we switch on the green channel, and within 120 s, the mixer chamber

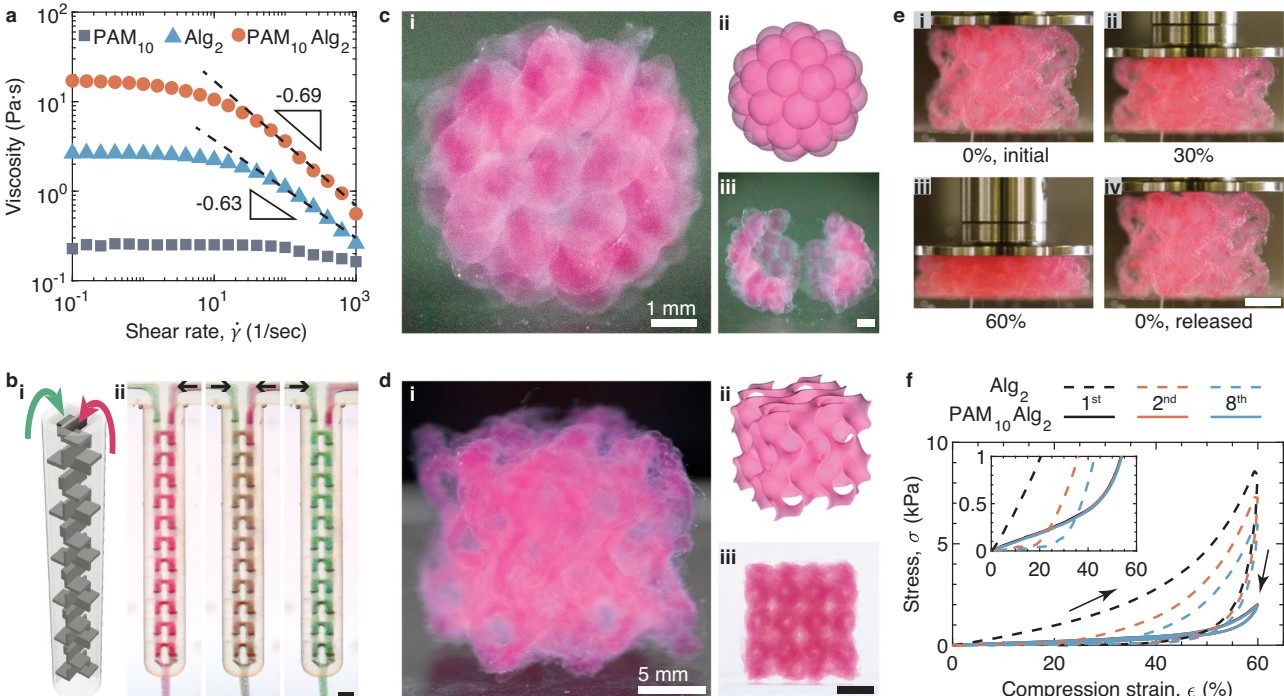

**Fig. 4 | 3D printing of DN hydrogels. a** The dependence of viscosity of the PAM (10%, w/v), Alg (2%, w/v), and the corresponding DN bio-ink, $PAM_{10}Alg_2$, on the shear rate, $\dot{\gamma}$, at RT. **b** A dual-inlet print nozzle with a static mixer chamber. (i) A 3D rendering of the static mixer chamber. (ii) Photos of the nozzle with a pair of DN $PAM_{10}Alg_2$ bio-inks flowing through the mixer chamber. The two bio-inks contain red and green microparticles, respectively, with a diameter of 50 μm. The left, middle, and right panels of (ii) are sequential photos when only the red channel flows, both channels flow, and only the green channel flows, respectively. Scale bar, 1 mm. **c** A DASP printed hollow sphere consisting of 42 interconnected yet distinguishable DN $PAM_{10}Alg_2$ hydrogel particles. Panels (i), (ii), and (iii) display the front view, computer-rendered model, and the hollow sphere that is cut into two pieces. Scale bar for panel (iii) is 1 mm. **d** A gyroid structure created by conventional printing that uses DN $PAM_{10}Alg_2$ hydrogel filaments as building blocks. Panels (i), (ii), and (iii) respectively depict the 45° view, computer-rendered 3D model, and side view of the gyroid. Scale bar for panel (iii) is 5 mm. **e** Photos of the gyroid structure from (**d**) undergoing cyclic compression testing with a compression strain rate of 0.005/s. Panels (i), (ii), (iii), and (iv) are captured at 0%, 30%, 60% compression strain, and full release in the first compression cycle. Scale bar, 5 mm. **f** Mechanical behavior of the gyroid structures during cyclic compression. The gyroids are made of DN $PAM_{10}Alg_2$ (solid lines) and single-network $Alg_2$ (dashed lines) hydrogels. Insert: a zoom-in view of the compression curves.

transitions to a greyish purple color because of thorough mixing (middle, Fig. 4b, ii). During this mixing process, approximately 36 μL of bio-ink passes through the chamber, which is around the volume of the chamber itself. Finally, we turn off the red channel, and within another 240 s, the chamber completely changes to a green color (right, Fig. 4b, iii). The smooth transition of colors in the mixer chamber demonstrates the nozzle's capability to mix the viscoelastic bio-inks homogeneously (Supplementary Movie 2).

To showcase the capabilities of DASP 2.0, we seek to fabricate a structure reminiscent of a raspberry: A hollow sphere with a shell composed of a single layer of hydrogel particles. Achieving this goal perhaps represents one of the most formidable tasks in voxelated bioprinting, as it demands precise generation, deposition, and assembly of individual particles in 3D space. Furthermore, it requires a robust connection between neighboring hydrogel particles to ensure the mechanical integrity of the structure.

Building on our previously established knowledge and leveraging the newly developed extrusion module, the printed droplets in the supporting matrix undergo isotropic swelling by 14% in diameter (Supplementary Fig. 11), enabling them to coalesce into a hollow sphere consisting of only one layer of interconnected yet distinguishable DN hydrogel particles. This sphere has a diameter of 7 mm and comprises 42 hydrogel particles, each approximately 1 mm in diameter, as shown by a representative photo in Fig. 4c, i and illustrated by the 3D rendering in Fig. 4c, ii. All the particles are interconnected yet distinguishable, resulting in a mechanically robust free-standing hollow sphere that can be easily manipulated without

additional protection (Supplementary Movie 3). Furthermore, we confirm the hollow nature of the sphere by cutting it into two pieces, revealing a single layer of hydrogel particles (Fig. 4c, iii; Supplementary Movie 4). Compared with the simple cubic lattice structure printed in our previous work[8], fabricating such a hollow sphere with high surface curvature represents a greater challenge. This is primarily due to the risk of the structure collapsing if the hydrogel itself is mechanically brittle. Therefore, the successful fabrication of this hollow sphere using DASP 2.0 represents a significant advancement in voxelated bioprinting.

In a DASP-printed structure, the bridges between neighboring droplets may be weaker than the bulk hydrogel, which would impair the mechanical robustness of the structure. To this end, we quantify the extensibility of a DASP-printed 1D filament that comprises a sequence of interconnected droplets (Supplementary Fig. 12). As expected, the 1D filament has a tensile breaking strain $\epsilon_f$ of 0.7, which is slightly lower than that of bulk hydrogel of 1.0 (Fig. 3c, d). Nevertheless, the breaking strain of the 1D filament remains much larger than the constituent single-network hydrogels ($\epsilon_f \approx 0.36$). These results further support that DASP enables the fabrication of integrated structures consisting of interconnected droplets.

DASP 2.0 can be easily converted to conventional one-dimensional (1D) filament-based bioprinting, as demonstrated by the creation of a gyroid cubic structure (Fig. 4d). Unlike other mesh-like structures such as waffles or honeycombs in which the channels are separated by walls, the gyroid is a highly porous structure consisting of an interconnected network of channels and voids (Fig. 4d, ii). To 3D

print such a complex geometric pattern, each layer must slightly protrude beyond the previous layer, forming the so-called hangovers. This poses a challenge for 3D bioprinting, as bio-inks often lack the mechanical strength to support the subsequent hangover layer. However, this caveat can be circumvented using our DN bio-ink, which not only is mechanically strong but also possesses relatively fast crosslinking kinetics. Indeed, with DASP 2.0, we successfully print our DN bio-ink into a gyroid with high fidelity, as confirmed by the 3×4 array of tunnel-like holes with a diameter of 1.5 mm on the side view (Fig. 4d, i & iii), precisely capturing the intended design (Fig. 4d, ii; Supplementary Movie 5).

To assess the mechanical properties of the DN gyroid, we perform a cyclic compression test at a strain rate of 0.005/s. The DN gyroid can sustain a large compression strain of 60% without fracturing (Fig. 4e, i-iii). Upon stress release, it completely recovers its original height, as visually depicted in Fig. 4e, iv and Supplementary Movie 6. The stress-strain profiles during loading and unloading almost overlap (solid lines, Fig. 4f). Quantitatively, the energy dissipation efficiency, defined as the ratio between the integrated area in the hysteresis loop and that under the compression curve, is relatively small with a value of 27.5%. This elastic, non-dissipative behavior persists over eight cycles of cyclic compression, as evidenced by the nearly perfectly overlaying of stress-strain profiles (solid lines, Fig. 4f). In contrast, the gyroid made of pure alginate exhibits plastic, dissipative behavior with the energy dissipation efficiency remarkably high of 80.2% at the first cycle. After eight compression cycles, the height of the gyroid decreases to 74% of its original value, as indicated by the increase of onset strain above which the stress becomes positive (dashed line, inset of Fig. 4f). Moreover, as the number of compression cycles increase to eight, the stress at 60% strain decreases by approximately 30% from 8.6 kPa to 6.0 kPa. These results demonstrate that DASP printing DN bio-inks enables the fabrication of highly deformable, elastic, non-dissipative, and multiscale porous scaffolds.

## Cytocompatibility and in vivo stability of DASP printed DN scaffolds

Compared to traditional scaffolds based on bulky hydrogels, the DASP-printed DN scaffolds offer a promising platform for cell-based therapy, which often involves encapsulating therapeutic cells for transplantation. The multiscale porosity has previously been shown to facilitate efficient nutrient transport[8], while the mechanical robustness is expected to allow the scaffolds to withstand constant mechanical perturbation resulting from the movement of the transplant recipient. To demonstrate this potential, we begin by investigating the cytocompatibility of both the printing process and the bio-ink by encapsulating cells into each hydrogel droplet. To do so, we modify the print nozzle by adding an extra channel, creating a three-channel nozzle (Supplementary Figs. 10b, c). Similar to the original dual-inlet print nozzle, two channels are used to load the DN ink. However, the third channel is dedicated to loading cells, as illustrated in Fig. 5a, i. This design ensures that the cells are not loaded directly to DN ink, as this could change the rheological properties of the ink. Such alterations can be significant and unpredictable at high cell density, rendering the ink unsuitable for DASP printing. Additionally, following our previously developed method[25], we incorporate 15% (w/v) dextran and 15% (w/v) poly(ethylene glycol) (PEG) polymers into the cell suspension and the DN ink, respectively. Compared to the DN ink, these polymer solutions are of relatively low viscosity and do not impact the printing process. This method results in an aqueous two-phase system, in which both the dextran and PEG solutions are aqueous and cytocompatible yet have a small interfacial tension that is sufficient to lead to phase separation[26]. During the extrusion process, the DN ink encounters the dextran phase in the static mixer chamber (Fig. 5a, i). After extrusion, the dextran phase is expected to phase separate from DN ink, forming an island-sea-like morphology,

as illustrated in Fig. 5a, ii. Indeed, this microstructure is confirmed by visualizing the distribution of dextran phase and cell-mimicking polystyrene microspheres with a diameter of 20 μm, respectively shown by a representative fluorescence confocal microscopy 3D image in Supplementary Fig. 13 and a bright field microscopy image in Fig. 5a, iii (see Methods). These results demonstrate that, together with the aqueous two-phase system, the three-channel nozzle allows for transforming the DN ink into a sponge-like structure with a solid hydrogel filled with cell-laden liquid voids.

By replacing the cell-mimicking microspheres with Beta-TC-6 cells, we print a 5×5×4 DN lattice (Supplementary Fig. 14). Beta-TC-6 cells are a pancreatic beta cell line that retains the function of glucose-stimulated insulin secretion, allowing us to test both the cell viability and the transport properties of the multiscale porous scaffold. The live/dead assay reveals that the viability of Beta-TC-6 cells immediately after printing is $61.3 \pm 8.0\%$, and it remains above 60% for the subsequent three days (Fig. 5b, c). This viability is consistent with the values 40-80% reported for extrusion-based bioprinting[27], but it is considerably lower compared to DASP 1.0, which uses pure alginate as the hydrogel and exhibits a viability of 90%[8]. The lower viability can be attributed to the cytotoxicity of free polyacrylamide polymers present in the DN ink before complete crosslinking. Although polyacrylamide-based hydrogels are generally considered as non-toxic and extensively used in biomedical research, our study reveals that free poly-acrylamide polymers exhibit some degree of cytotoxicity. Indeed, live/dead assay conducted on cells directly mixed with pure poly-acrylamide polymers with the same concentration as that used for the DN bio-ink shows a cell viability less than 1% (Supplementary Fig. 15). Nevertheless, in DASP 2.0 more than half of the cells survive and maintain their viability for three days, which is a typical waiting time before transplantation in clinical settings.

Next, we test the insulin release of the lattice scaffolds using static glucose-stimulated insulin secretion (GSIS) measurement. Specifically, we expose the Beta-TC-6 cell-encapsulated scaffolds to low and high glucose media and measure the amount of released insulin. The release index is defined as the ratio between insulin release amount in high glucose media over that in low glucose media. On the first day, the lattice scaffolds exhibit a relatively low release index with a value of $1.39 \pm 0.56$. Yet, this value is comparable to that of the naked counterpart, which exhibits a release index of $1.31 \pm 0.42$ (Fig. 5d). This is likely because that the cells are in a recovery phase after detachment from the culture flask. In the subsequent days, the release index of the naked cells increases significantly, reaching a value of $6.78 \pm 1.03$ on day 3, which is about twice the value of $2.98 \pm 0.83$ observed for the lattice scaffolds (Fig. 5d). This difference is consistent with our previous studies, in which we found that the release index of islets encapsulated 3×3×3 lattice scaffolds created using DASP 1.0 is about half that of the naked counterpart. Nonetheless, the release index of DASP Beta-TC-6 cells on day 3 is much higher than one, which exceeds the standard threshold for clinical transplantation. Taken together, our results show that DASP 2.0 enables the encapsulation of living cells while maintaining reasonable cell viability and function.

To further confirm the suitability of the DN scaffolds for clinical transplantation, we perform in vivo studies to evaluate their mechanical robustness. We transplant the 5×5×4 lattice scaffolds into the abdominal cavity of immunocompetent C57BL/6 mice, as respectively illustrated and photographed in Fig. 5e, i and ii. This in vivo environment is characterized by vigorous mechanical perturbation resulting from intestinal peristalsis and body movement. Two bio-ink formulations are used to prepare the scaffolds: (i) 4% (w/v) alginate, our previous prototype formulation for DASP 1.0, denoted as $Alg_4$, and (ii) DN bio-ink for DASP 2.0, a formulation explored for cell encapsulation, denoted as $PAM_{10}Alg_2$ (Fig. 5a). Both $Alg_4$ and $PAM_{10}Alg_2$ scaffolds exhibit similar stiffness, with respective apparent compression moduli of 5.6 and 5.8 kPa (dashed red line and dark line, Fig. 5h). However, a

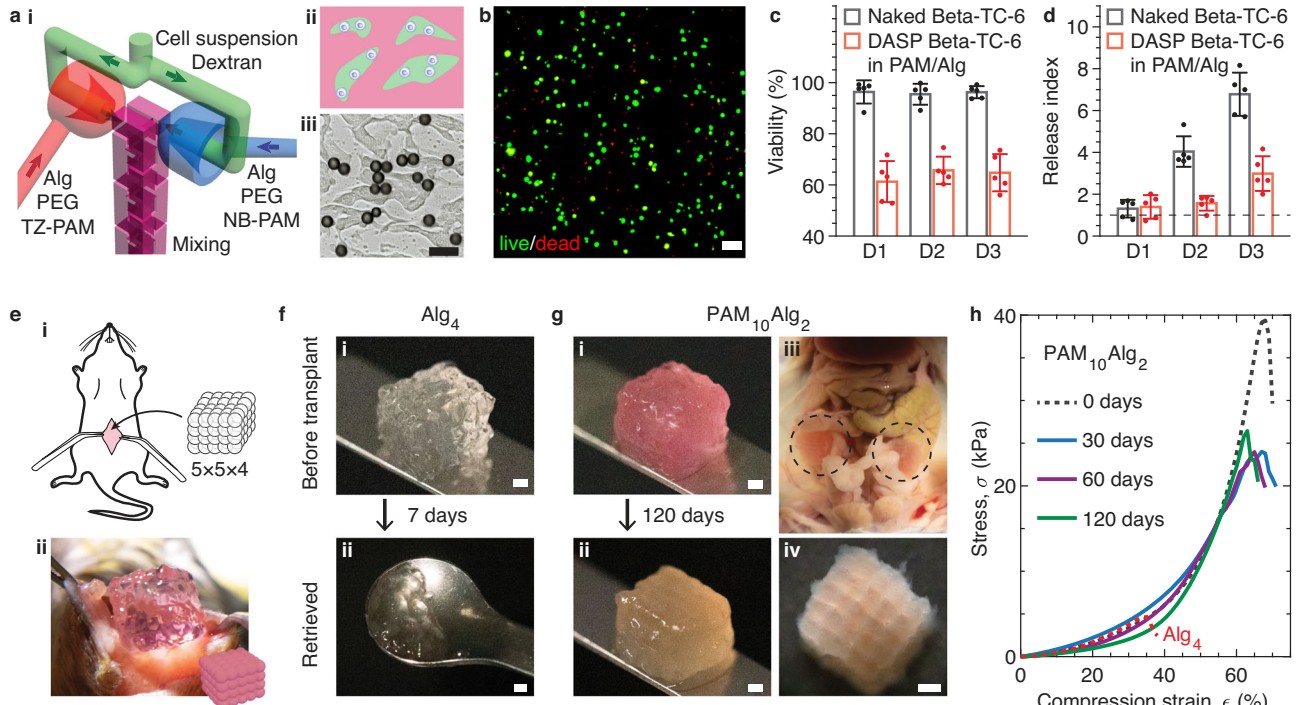

**Fig. 5 | Cytocompatibility and in vivo stability of DASP printed DN polyacrylamide/alginate (PAM/Alg) scaffolds.** The single-network and DN scaffolds are respectively made of $Alg_4$ and $PAM_{10}Alg_2$ hydrogels. **a** A strategy that utilizes PEG/dextran aqueous two-phase system to encapsulate cells during DASP printing. (i) A 3D rendering of the print nozzle for cell encapsulation. The print nozzle consists of three channels: one channel loaded with cell suspension in dextran (green flow) and two channels respectively loaded with the pair of bio-inks mixed with PEG (red and blue flow). During the extrusion, the flows of bio-ink first encounter with the immiscible flow of the cell suspension and subsequently mix in the static mixer chamber. (ii) A schematic of the extruded bio-ink in which the cell suspension phase (green) is separated from the bio-ink phase (red). (iii) A representative microscopy image showing the distribution of cell-mimicking polystyrene microparticles in the extruded bio-ink. Scale bar, 50 μm. **b** A representative fluorescence confocal microscopy image from live/dead assay of DASP encapsulated Beta-TC-6 cells. Scale bar, 100 μm. **c** Viability of naked and DASP encapsulated Beta-TC-6 cells up to 3 days. **d** Glucose stimulation index of naked and DASP Beta-TC-6 cells in 3 days. Results in (**c**) and (**d**) are shown as mean ± standard deviation (S.D.) with sample size $n = 5$ wells for naked Beta-TC-6 and $n = 5$ scaffolds for DASP Beta-TC-6 in PAM/Alg. **e** An animal model for testing the mechanical robustness of DASP printed DN scaffolds in vivo. (i) A schematic of transplanting a DASP printed 5×5×4 lattice scaffold into the abdominal cavity of immunocompetent C57BL/6 mice. (ii) A representative photograph of the surgery. **f** Representative photos of the scaffolds made of pure alginate hydrogel (i) before the transplantation and (ii) after being retrieved. Scale bars, 1 mm. **g** Representative photographs of the scaffolds made of DN hydrogel (i) before the transplantation, (ii) after being retrieved, (iii) located in the abdominal cavity (dashed circles), and (iv) under digital camera after being retrieved. Scale bars, 1 mm. **h** Mechanical behavior of scaffolds retrieved at different time points under compression.

significant difference in in vivo stability is observed for these two scaffolds. $Alg_4$ scaffolds fragment only 7 days after the transplantation and can be hardly retrieved, as shown in Fig. 5f. This phenomenon can be attributed to the fragility of the alginate scaffold, which fractures under a relatively small strain of 37% during compression test (dashed red line, Fig. 5h), and limited stability of ionic crosslinks in physiological conditions[28]. In contrast, the $PAM_{10}Alg_2$ scaffolds retain their integrity without observable deformation for 120 days after the transplantation, as shown by Fig. 5g, i and ii. As observed in different mice, the $PAM_{10}Alg_2$ scaffolds are widely distributed in the abdominal cavity, in contact with the stomach, intestine (Fig. 5g, iii), bladder, liver (Supplementary Fig. 16a), and sometimes slightly adhering to fat tissue, yet all scaffolds could be successfully retrieved (Supplementary Fig. 16b). Moreover, the retrieved scaffolds maintain the pre-transplant lattice structure in which individual particles are interconnected yet distinguishable (Fig. 5g, iv).

The integrity of the retrieved DN scaffolds is further confirmed by their mechanical robustness. For scaffolds retrieved 30 days after transplantation, their yield compression stress decreases from 39.3 kPa to 23.8 kPa compared to its pre-transplant value; however, the yield compression strain remains remarkably high of approximately 60% (solid blue line, Fig. 5h). Moreover, the stress-strain curve nearly perfectly overlaps with that of the pre-transplant one, and this behavior persists for scaffolds retrieved after longer periods of

transplantation for 60 and 120 days (purple and green lines, Fig. 5h). Taken together, our results show that the DASP 2.0 printed DN scaffolds possess sufficient mechanically robustness to maintain long-term in vivo stability.

**Universality of DN (A + B)/C hydrogels as bio-inks for DASP 2.0**
We note that the cytocompatibility of the PAM/alginate DN bio-ink is relatively low, with cell viability ~60% (Fig. 5c), largely because free PAM-based polymers are not completely cytocompatible (Supplementary Fig. 15). Yet, the concept of (A + B)/C DN hydrogels based on TZ-NB click chemistry is expected to be applicable to other biopolymers, where A, B, and C respectively represent TZ-polymer, NB-polymer, and alginate. To test this, we replace the PAM polymer by hyaluronic acid (HA), a natural polysaccharide widely used in biomaterials development. Following the same chemistry as for the PAM (Fig. 1b), we functionalize a pair of HA polymers, respectively, with TZ and NB at nearly the same grafting ratio of 27%, corresponding to approximately 250 TZ or NB groups per polymer (Supplementary Figs. 6, 7). Upon mixing, the pair of TZ-HA and NB-HA crosslink to form a network via additive-free click reaction, as illustrated in Fig. 6a.

Compared to the single-network hydrogels, the improvement in the mechanical properties of the HA/alginate DN hydrogel is comparable to that observed for the PAM/alginate formulation. For

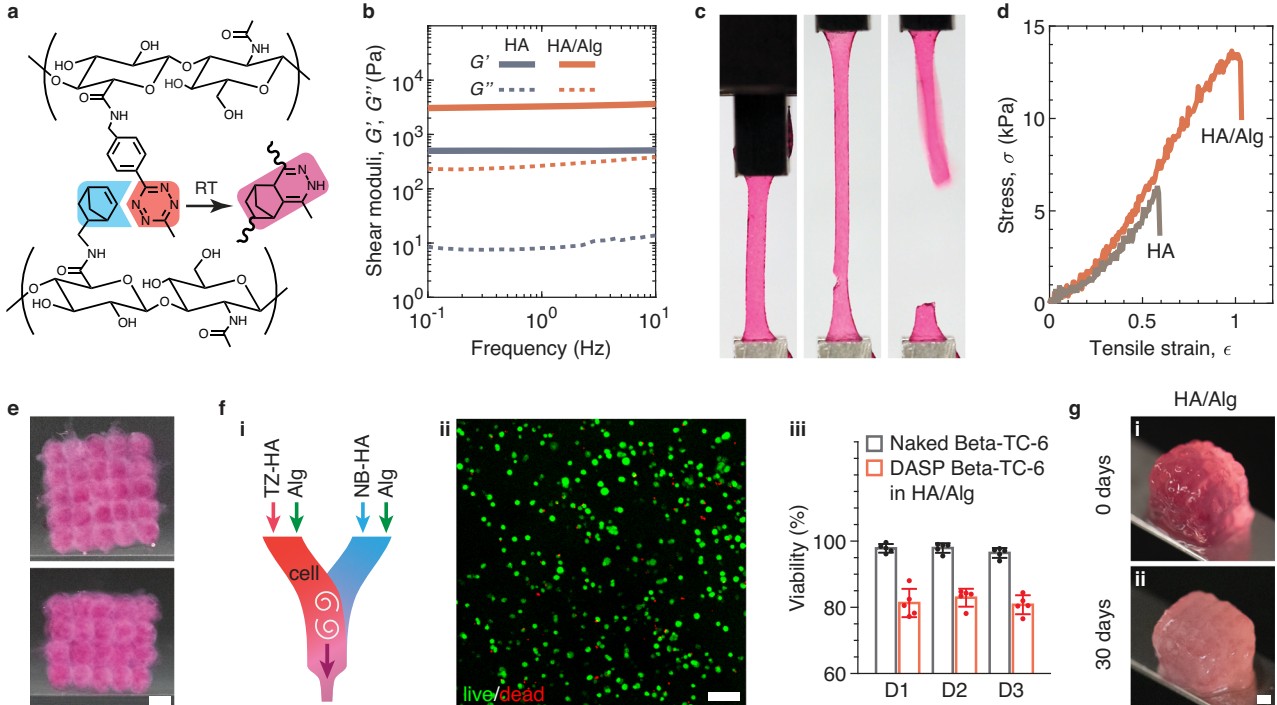

**Fig. 6 | Mechanical properties, printability, and biocompatibility of a DN hyaluronic acid/alginate (HA/Alg) bio-ink. a** HA is functionalized with either norbornene (NB) or tetrazine (TZ) groups. Upon mixing, TZ-HA and NB-HA are crosslinked via the click reaction between TZ and NB groups, following the same crosslinking mechanism as that for creating a PAM network (Fig. 1b). The grafting ratio of TZ-HA and NB-HA are both fixed at 27%. **b** Dependencies of storage ($G'$, solid lines) and loss ($G''$, dashed lines) moduli of completely crosslinked single-network HA and DN HA/alginate hydrogels on oscillatory shear frequency. Single-network hydrogel (HA): TZ-HA (2%, w/v) + NB-HA (4%, w/v). The DN hydrogel (HA/alginate) is prepared by mixing a pair of bio-inks, TZ-HA (2%, w/v) / alginate (2%, w/v) + NB-HA (4%, w/v) / alginate (2%, w/v). **c** Representative photos of the DN HA/alginate hydrogel under uniaxial tensile test at a fixed strain rate of 0.02/s. The left, middle, and right panels, respectively, are captured at the beginning of the tensile test, right before fracture, and right after fracture. Samples are 13 mm in length and 2 mm in width of the center part. **d** Stress-strain behavior of the single-network HA and DN HA/alginate hydrogels in (**b**). **e** A DASP printed 5×5×4 lattice scaffold consisting of 100 interconnected yet distinguishable DN HA/alginate hydrogel particles. Upper panel: top view; lower panel: side view. Scale bar, 1 mm. **f** Cytocompatibility of the DN HA/alginate bio-ink. (i) A pair of HA/alginate bio-inks pre-loaded with cells are printed using a two-channel print nozzle. (ii) A representative fluorescence confocal microscopy image from live/dead assay of DASP encapsulated Beta-TC-6 cells within the DN HA/alginate hydrogel. Scale bar, 100 μm. (iii) Viability of naked and DASP encapsulated Beta-TC-6 cells within the DN HA/alginate hydrogel up to 3 days. Results are shown as mean ± standard deviation (S.D.) with sample size $n = 5$ wells for naked Beta-TC-6 and $n = 5$ scaffolds for DASP Beta-TC-6 in HA/Alginate. **g** Representative photographs of the scaffolds made of DN HA/alginate hydrogel (i) before the transplantation, (ii) after being retrieved at 30 days. Scale bar, 1 mm.

instance, the shear modulus of a single-network HA is about 500 Pa (solid gray line, Fig. 6b). By contrast, for the DN HA/alginate hydrogel, $G$ is about 3050 Pa, approximately the sum of HA and alginate single-network hydrogels (red solid line, Fig. 6b). Moreover, unlike the single-network HA hydrogel that is brittle with a tensile breaking strain of 0.6, the DN HA/alginate hydrogel is significantly more stretchable with $\epsilon_f = 1.02$, as shown in Fig. 6c, d and Supplementary Movie 7.

As expected, the HA/alginate is suitable for DASP printing, as demonstrated by the success of creating a 5×5×4 lattice scaffold (Fig. 6e). To test the cytocompatibility of the HA/alginate hydrogel, we directly mix Beta-TC-6 cells with the bio-ink at a cell density of 5 million/mL. The live/dead assay reveals that the viability of HA/alginate encapsulated Beta-TC-6 cells remains above 80% for three days after the printing (Fig. 6f). This survival rate is significantly higher than 60% observed in PAM$_{10}$Alg$_2$, and importantly, meets the standards for most applications[27]. Finally, we transplant the DASP printed 5×5×4 lattice scaffolds made of HA/alginate into the mouse abdominal cavity. The integrity of the retrieved scaffolds is confirmed 30 days after transplantation (Fig. 6g), indicating that HA/alginate DN scaffolds are sufficiently robust to maintain long-term in vivo stability. Taken together, these results demonstrate that the concept of (A + B)/C hydrogels provides a universal strategy for the development of modular DN bio-inks for DASP printing.

## Discussion

The advancement of voxelated bioprinting requires three indispensable efforts: (i) biomaterial voxels, (ii) technology development, and (iii) demonstration of biomedical applications. Through a combination of biomaterials development and hardware engineering, we have significantly advanced our voxelated bioprinting technology from DASP 1.0 to DASP 2.0. Compared to DASP 1.0 that can print pure alginate voxels only, DASP 2.0 allows for printing DN hydrogel voxels consisting of interpenetrating alginate and PAM or HA polymers. Unlike classical DN PAM/alginate hydrogels crosslinked by cytotoxic free radical polymerization of acrylamide monomers[22], our DN hydrogel is formed by crosslinking PAM or HA polymers through a biofriendly click chemistry. This is achieved by controlled synthesis of a pair of precursor PAM or HA polymers, which are functionalized with prescribed fractions of tetrazine and norbornene groups, respectively. Upon mixing, a pair of tetrazine and norbornene groups forms a covalent bond through click reaction without any external trigger. In the meantime, the alginate network is formed by ionic crosslinking. These two kinds of crosslinking mechanisms are orthogonal, allowing for modular control over hydrogel mechanical properties through its constituent polymer networks. Further, we engineer multi-channel print nozzles to enable on-demand mixing of highly viscoelastic bio-inks without significantly impairing cell viability. Exploiting the print nozzle allows for the fabrication of highly complex 3D structures such

as a hollow sphere made of one layer of interconnected yet distinguishable hydrogel particles. Finally, we conduct in vitro studies to show that voxel-printed DN scaffolds are cytocompatible, as well as in vivo studies to demonstrate that the scaffolds are mechanically robust to be easily retrieved after being transplanted into immunocompetent mice.

The concept of using a pair of stoichiometrically matched functionalized polymers for hydrogel synthesis offers precise control over crosslinking kinetics and stiffness. In classical HA[29] and PEG[30] based hydrogels, multiple-functional polymers are pre-mixed with small molecule crosslinkers, which upon external triggers such as light crosslink the polymers to form a network. As a result, the gelation kinetics is often sensitive to the type and conditions of chemical reactions and can be difficult to be precisely controlled. By contrast, our hydrogels are crosslinked by click reaction, a process that is relatively insensitive to reaction conditions. The gelation kinetics can be well-described by a one-parameter theory accounting the concentration of crosslinkers. Moreover, the hydrogel stiffness increases with the polymer concentration by a power of 2.1; this relation is well-described by a simple polymer physics model that the network stiffness is $k_B T$ per correlation volume of the polymer solution. Compared to the linear dependence of stiffness on crosslinker concentration in conventional hydrogels, this power law behavior offers a wider range of tunable hydrogel stiffness.

Unlike single-network hydrogels whose stiffness and extensibility are intrinsically negatively correlated, the developed DN bio-ink offers modular control over mechanical properties. For example, the stiffness of the DN hydrogel is about the sum of the moduli of its constituent single-network hydrogels. In contrast, the elongation at break is determined by the size of PAM network strand. Consequently, one can control the extensibility of a DN hydrogel by varying the crosslinking density of the PAM network and control the stiffness by varying the concentration of alginate. Such a modular control over the mechanical properties may offer an approach to designing tough biomaterials for applications such as engineering blood vessels[31] and cartilage repair[32].

In the context of hardware engineering, the capability of integrating multiple flows into a single print nozzle offers versatility in processing bio-inks. For instance, the developed dual-inlet print nozzle allows for mixing a pair of highly viscoelastic biomaterials on demand. The design of the nozzle is inspired by the commercially available static mixer Quadro™ Square. Yet, leveraging the capability of stereolithography printing, the dimension of the print nozzle is dramatically reduced to ensure a minimal dead volume; this allows us to use a relatively small amount of bio-ink less than 1 mL during processing including but not limited to bioprinting. An example is using this nozzle to rapidly extrude mixtures of A + B hydrogels onto rheometer for real-time characterization of crosslinking kinetics and dynamic mechanical properties.

The developed triple-inlet print nozzle allows for leveraging the existing aqueous two-phase system[26] to encapsulate cells at high density without significantly impairing cell viability. Moreover, the printed hydrogel voxels possess a multiscale porous structure, in which cell containing microscale liquid domains are randomly dispersed into a solid DN hydrogel. This structure may offer advantages in efficient nutrient transport essential to cell survival[33]. We note that free PAM polymers are not suitable for long-term cell culture because of limited cytocompatibility (Supplementary Fig. 15). Although this caveat can be mitigated using the triplet-inlet print nozzle to avoid directly mixing the cells with the PAM polymers for a long time, the cytocompatibility of the bio-ink remains relatively low with a cell viability of 60%. However, this issue can be solved by replacing the PAM backbone with a HA polymer, which is cytocompatible and can be directly pre-mixed with cells during the printing without significantly impairing cell viability.

The success of voxelated printing DN bio-inks relies on on-demand mixing, extruding, and crosslinking A + B type biomaterials without external trigger. Such a capability is enabled by integrating additive-free click chemistry and a multi-channel print nozzle. This design strategy has important implications in embedded bioprinting, in which the deposited bio-inks are trapped inside a thick supporting matrix. The matrix prevents easy access to external triggers such as UV or heat, which are often required to solidify the printed structures[34]. In our printing process, the additive-free click chemistry allows the bio-inks to be solidified before removing the supporting matrix, enabling the fabrication of highly complex 3D objects with high fidelity (Fig. 4c, d). This capability may be appealing for administering biomaterials in vivo, which is an environment with limited access to external triggers. Finally, the developed multi-channel print nozzles can be easily adapted to medical syringes and needles for administering the DN bio-inks as a class of injectable biomaterials[35].

We note a few limitations and potential challenges associated with applying DASP 2.0 for basic and translational biomedicine. For instance, to exploit DASP 2.0 to understand and control cell-matrix interactions, it is necessary to render the DN hydrogels cell-instructive. To this end, when preparing the DN HA/alginate hydrogel, the TZ-HA and NB-HA are mixed with a mismatched NB/TZ ratio of 2/1 (Fig. 6a), resulting in considerable unreacted free NB groups in the hydrogel network. These free NB groups may be used to conjugate extracellular matrix-derived peptides for cell engagement[36,37]. Moreover, degradable crosslinkers[38,39] and growth factors[40] may be conjugated into the hydrogel network to promote cell migration and vascularization.

In the context of translational biomedicine, our results highlight the potential of voxelated bioprinting in manufacturing cell-encapsulated multiscale porous and mechanically robust scaffolds as therapeutic transplants[41,42]. For instance, a potential application is to encapsulate human islets in a voxelated 3D scaffold, which may be used as a transplant to reverse type 1 diabetes. Since a voxelated 3D scaffold consists of interconnected yet distinguishable hydrogel particles, it may combine the advantages of classical microencapsulation technology in high surface-to-volume ratio[43,44] and macro-devices in easy retrievability[45,46]. Yet, it remains to be determined the biomaterial biocompatibility and the ability of voxelated 3D scaffolds to allow long-term cell survival and function. Nevertheless, DASP 2.0 represents a significantly advanced voxelated bioprinting technology, paving the way for engineering highly complex yet organized functional 3D tissue constructs.

## Methods
### Materials and reagents for chemical synthesis, bioprinting, and cell culture
**Chemical synthesis.** 4-(Aminomethyl)benzonitrile hydrochloride (Cat. No. 631396), triethylamine (Cat. No. 471283), di-*tert*-butyl dicarbonate (Boc₂O, Cat. No. 205249), hydrazine (anhydrous, Cat. No. 215155), trifluoroacetic acid (Cat. No. T6508), poly(acrylamide-co-acrylic acid) partial sodium salt (MW 150 kDa, acrylamide ~80 wt. %, Cat. No. 511471) are purchased from Sigma-Aldrich, USA. Nickel(II) trifluoromethanesulfonate (Ni(OTf)₂, Cat. No. N08615G), 4-(4,6-dimethoxy-1,3,5-triazin-2-yl)-4-methylmorpholinium chloride (DMTMM, Cat. No. D29195G), and 5-norbornene-2-methylamine (Cat. No. N09075G) are purchased from Tokyo Chemical Industry (TCI), Fisher Scientific, USA.

**Voxelated bioprinting.** Gelatin from porcine skin (gel strength 300, Type A, Cat. No. G2500), alginic acid sodium salt from brown algae (medium viscosity, Cat. No. A2033), dextran from *Leuconostoc* spp. with MW approximately 40 kDa (Cat. No. 31389), polystyrene microparticles with a diameter of 20 μm (Cat. No. 74491), and fluorescein isothiocyanate−dextran (2000 kDa, Cat. No. FD2000S) are purchased from Sigma-Aldrich, USA. Poly(ethylene glycol) (PEG) with MW of

8 kDa (Cat. No. 43443) is purchased from Fisher Scientific, USA. Red and green mica powders for visualization of bio-ink flows are purchased from SEISSO, China.

**Cell culture and characterization.** Beta-TC-6 cells are received as a gift from Dr. Yong Wang, which was purchased from the American Type Culture Collection (Cat. No. CRL-11506). Dulbecco's Modified Eagle Medium (DMEM, high glucose, GlutaMax, Cat. No. 10566-016), DMEM (high glucose, without calcium, Cat. No. 21068-028), FBS (Cat. No. 10438-018), Pen/Strep (Cat. No. 15140-122), 2-mercaptoethanol (Cat. No. 21985- 023) are purchased from Gibco, Fisher Scientific, USA. Chemical dye for live/dead assay including fluorescein diacetate (Cat. No. F7378), propidium iodide (Cat. No. P4170) are purchased from Sigma-Aldrich, USA.

### Chemical and polymer synthesis

**Compound I (CI): tert-butyl N-(4-cyanobenzyl) carbamate.** CI is prepared following a previously reported procedure[47] with minor modifications. To a solution of 4-(aminomethyl)benzonitrile hydrochloride (10.00 g, 59.3 mmol) in dichloromethane (250 mL), triethylamine (20.7 mL, 148.5 mmol) and Boc$_2$O (14.23 g, 65.2 mmol) are gently added. The reaction mixture is stirred at RT for 12 h. The resulting solution is washed with 10% HCl (200 mL) to remove the excessive triethylamine and unreacted 4-(aminomethyl)benzonitrile hydrochloride if any. Finally, DCM is removed in vacuo. The obtained product is a white solid with 98% yield. The product is verified by $^1$H-NMR (Supplementary Fig. 1) and used for the synthesis of CII without further purification.

**Compound II (CII): tert-butyl *N*-(4-(6-methyl-1,2,4,5-tetrazin-3-yl) benzyl)carbamate.** CII is prepared following a previously reported procedure[48]. CI (2.00 g, 8.6 mmol), acetonitrile (4.5 mL, 86.2 mmol), Ni(OTf)$_2$ (1.53 g, 4.3 mmol), and hydrazine (13.8 mL, 430.6 mmol) are added to a 60 mL high-pressure tube (ACE GLASS, Sigma-Aldrich, Cat. No. Z568872). The tube is sealed, and the reaction mixture is stirred at 60 °C for 24 h. The resulting mixture is diluted with EtOAc (200 mL), transferred to a separatory funnel, and washed with brine (100 mL) for three times to remove the excessive hydrazine and acetonitrile. The intermediate is further oxidized into CII following a previously reported method[48]. Finally, EtOAc is removed in vacuo and the crude product is obtained as dark purple solid. The raw product is further purified by silica gel column chromatography using gradient elution with EtOAc/Hexane from 8:92 to 20:80. The yield for the synthesis of CII is 59%. The product is verified by $^1$H-NMR (Supplementary Fig. 2).

**Compound III (CIII): 4-(6-Methyl-1,2,4,5-tetrazin-3-yl)benzenemethanamine.** To a solution of CII (5 g, 16.6 mmol) in DCM (25.4 mL), trifluoroacetic acid (25.4 mL, 332 mmol) is added and the reaction mixture is stirred at RT for 2 h. Then, DCM is removed in vacuo and saturated NaHCO$_3$ solution is added dropwise to neutralize the residual trifluoroacetic acid. At increased pH caused by NaHCO$_3$, most of CIII precipitates due to the transformation from the HCl salt state to the original state, yet there is a slight amount of CIII dissolved in the aqueous phase. Finally, the precipitated product and the minor portion in the aqueous phase are extracted with DCM. After the removal of DCM in vacuo, CIII is obtained as a pink solid with 99% yield. The product is verified by $^1$H-NMR (Supplementary Fig. 3) and used for modifying polymers without further purification.

**Tetrazine-modified poly(acrylamide-co-acrylic acid).** DMTMM (4.66 g, 16.8 mmol) is added to a solution of poly(acrylamide-*co*-acrylic acid) (PAmAc) (10 g, 140 mmol chemical monomer) in 300 mL mixed solvent of water/MeOH (2:1 v/v). The reaction mixture is stirred at RT for 30 min, then CIII (0.84 g, 4.2 mmol) suspended in 10 ml MeOH is added. The reaction mixture is further stirred at RT for 48 h. The

resulting polymer solution is precipitated in 1.5 L acetone twice. Then, the precipitated polymer is collected and redissolved in 200 mL water and dialyzed (Spectrum™, 3.5 kDa molecular weight cut-off (MWCO), Cat. No. 132112) against DI water for further purification. During dialysis, the outer phase is refreshed 3 times/day for 72 h. Finally, the polymer is dried through lyophilization. The grafting density measured by $^1$H-NMR (Supplementary Fig. 4) is 2.53%, which is defined as the molar ratio between the grafted tetrazine and the total number of chemical monomers on the PAmAc backbone.

**Norbornene-modified poly(acrylamide-co-acrylic acid).** DMTMM is added to a solution of PAmAc (10 g, 140 mmol chemical monomer) in 300 mL mixed solvent of water/MeOH (2:1 v/v). The reaction mixture is stirred at RT for 30 min, then 5-norbornene-2-methylamine dissolved in 10 ml MeOH is added and the mixture is further stirred at RT for 48 h.

To synthesize NB-PAmAc with various grafting density, we change the molar ratio of the feeding norbornene to the total number of chemical monomers on the polymer backbone. Specifically, 5-norbornene-2-methylamine is added as 0.35 g (2.8 mmol), 0.52 g (4.2 mmol), 0.69 g (5.6 mmol), and 1.04 g (8.5 mmol); whereas DMTMM is respectively added as 3.09 g (11.2 mmol), 4.64 g (16.8 mmol), 6.18 g (22.4 mmol), and 9.38 g (34.0 mmol).

The resulting polymer solution is precipitated in 1.5 L acetone twice. Then, the precipitated polymer is collected and redissolved in 200 mL water and dialyzed against DI water (Spectrum™, 3.5 kDa molecular weight cut-off (MWCO), Cat. No. 132112) for further purification. During dialysis, the outer phase is refreshed 3 times/day for 3 days. Finally, the product is dried through lyophilization. The grafting ratio measured by $^1$H-NMR (Supplementary Fig. 5) is 0.63%, 1.51%, 2.57%, and 4.25%, respectively.

**Tetrazine-modified sodium hyaluronate.** DMTMM (2.00 g, 7.2 mmol) is added to a solution of sodium hyaluronate (HA) (2 g, 5 mmol chemical monomer) in 200 mL mixed solvent of water/MeOH (2:1 v/v). The reaction mixture is stirred at RT for 30 min, then CIII (0.36 g, 1.8 mmol) suspended in 10 ml MeOH is added. The reaction mixture is further stirred at RT for 48 h. The resulting polymer solution is precipitated in 1 L ethanol for twice. Then, the precipitated polymer is collected and redissolved in 200 mL water and dialyzed (Spectrum™, 3.5 kDa molecular weight cut-off (MWCO), Cat. No. 132112) against DI water for further purification. During dialysis, the outer phase is refreshed 3 times/day for 72 h. Finally, the polymer is dried through lyophilization. The grafting density measured by $^1$H-NMR (Supplementary Fig. 6) is 27%, which is defined as the molar ratio between the grafted tetrazine and a total number of chemical monomers on the HA backbone.

**Norbornene-modified sodium hyaluronate.** DMTMM (2.66 g, 9.6 mmol) is added to a solution of sodium hyaluronate (HA) (2 g, 5 mmol chemical monomer) in 200 mL mixed solvent of water/MeOH (2:1 v/v). The reaction mixture is stirred at RT for 30 min, then 5-norbornene-2-methylamine (0.3 g, 2.4 mmol) suspended in 10 ml MeOH is added. The reaction mixture is further stirred at RT for 48 h. The resulting polymer solution is precipitated in 1 L ethanol twice. Then, the precipitated polymer is collected and redissolved in 200 mL water and dialyzed (Spectrum™, 3.5 kDa molecular weight cut-off (MWCO), Cat. No. 132112) against DI water for further purification. During dialysis, the outer phase is refreshed 3 times/day for 72 h. Finally, the polymer is dried through lyophilization. The grafting density measured by $^1$H-NMR (Supplementary Fig. 7) is 27%, which is defined as the molar ratio between the grafted norbornene and a total number of chemical monomers on the HA backbone.

### Instrument setup of DASP 2.0
Building on DASP 1.0[8], we develop the instrument of DASP 2.0 by improving the extrusion module and the print nozzle. First, we develop

an extrusion module for simultaneously loading and extruding up to three syringes (Supplementary Fig. 10a). Second, the print nozzle of DASP 2.0 is made of a dispensing needle (26 G, McMaster-Carr) and an upstream mixing module with an inner static mixer chamber, such that bio-inks can be mixed on-demand during extrusion. The mixing module is manufactured through stereolithography 3D printing (Formlabs 3 + ). To demonstrate printing DN bio-inks and cell-encapsulated DN scaffolds, we respectively design a dual-inlet and a triple-inlet print nozzle, as visualized in Supplementary Figs. 10b, c. The dual-inlet print nozzle allows a pair of bio-inks to be mixed in the mixer chamber. The triple-inlet print nozzle ensures that a pair of bio-inks meet the cell suspension before being mixed in the static mixer chamber. During printing, the print nozzle is attached to the extrusion module and connected with syringes (1 mL, Shanghai Bolige Industrial & Trade Co., Ltd) through tubing (Cat. No. BB31695-PE/5, Scientific Commodities Inc).

## Bio-inks

To explore the crosslinking kinetics and stiffness of single-network PAM hydrogels (Fig. 2), we prepare the bio-inks by respectively dissolving TZ-PAM and NB-PAM in DI water. This approach avoids possible effects from salt ions on the conformation of polymer chains, simplifying the corresponding theoretical calculations. In the subsequent study on the stiffness and extensibility of DN hydrogels (Fig. 3), we used 150 mM NaCl solution to respectively prepare PAM, alginate, and DN PAM/alginate bio-inks. Specifically, to prepare a DN bio-ink, we respectively dissolve TZ-PAM and NB-PAM in 150 mM NaCl solution, and then add the same amount of alginate to each of the two solutions. These bio-inks are isotonic and suitable for the following mechanical test in which the hydrogels are required to equilibrate in the cell culture medium. For cell encapsulation and animal studies, the concentrations of the TZ-PAM, NB-PAM, and alginate are respectively fixed at 10%, 10%, and 2% (w/v). Additionally, the NaCl solution is replaced by DI water with 15% (w/v) PEG and 40 mM HEPES.

To prepare the HA, alginate, and DN HA/alginate bio-inks in Fig. 6, we use 150 mM NaCl solution to prepare TZ-HA, NB-HA, and alginate bio-ink. Following the same protocol described in PAM/alginate bio-ink, we prepare HA/Alginate bio-ink with the concentration of TZ-HA, NB-HA, and alginate respectively fixed at 2%, 4%, and 2% (w/v). For cell encapsulation and animal studies, the NaCl solution is replaced by DMEM (without calcium) media supplemented with 30 mM HEPES.

## Rheometry

Rheological measurements are performed using a stress-controlled rheometer (Anton Paar MCR 301) equipped with a parallel-plate geometry of diameter 25 mm at 20 °C. To prevent the evaporation of water from hydrogels, we apply mineral oil to seal the edge of the geometry for all samples. The correction to the measured moduli from mineral oil is negligible according to our previous studies[8].

To prepare the sample for single-network PAM or HA hydrogels, we use dual-inlet print nozzle to manually extrude the bio-ink on the bottom geometry of the rheometer, lower the geometry to a 1 mm gap, and trim the edge of the liquid. We characterize the crosslinking kinetics by monitoring in real time the storage modulus $G'$ and loss modulus $G''$ at an oscillatory shear frequency of 1 Hz and a shear strain of 0.5% for 12 h. This duration is sufficient for complete crosslinking of the hydrogel. The time taken for mixing the bio-inks is included when determining crosslinking kinetics. Following the measurements on kinetics, we apply an oscillatory frequency shear on the crosslinked hydrogel to characterize its shear moduli. The frequency is changed from 0.1 to 100 Hz and the strain is fixed at 0.5 %. The resulting data, shown in Fig. 2d, encompasses a frequency range from 0.1 to 20 Hz due to sample slippage at higher frequencies. In parallel, we measure the shear moduli of single-network hydrogels in DMEM (Fig. 3a, b, e). Unlike DI water, DMEM contains salt ions that are known to slightly affect hydrogel mechanical properties. To this end, we

incubate a hydrogel in DMEM for 24 h to reach equilibrium before measurement.

To prepare the sample of a single-network alginate hydrogel, we prepare a film of alginate solution with an approximate thickness of 1 mm. To crosslink the film, we immerse it in a bath consisting of 20 mM $CaCl_2$ and 120 mM NaCl in DI water for 10 min. We then transfer the solidified film to DMEM supplemented with 2 mM $CaCl_2$ and incubate for 24 h to equilibrate the film. We cut the film into a disk with a diameter of 25 mm, matching the dimension of the rheometer geometry. The hydrogel disk is loaded onto the bottom geometry of the rheometer, and the upper geometry is lowered until it contacts the sample, which is indicated by a positive normal force of approximately 0.1 N. To characterize the shear moduli, an oscillatory shear is applied with a fixed strain of 0.5% and various frequency ranging from 0.1 to 100 Hz. A similar procedure is used to prepare and measure the dynamic mechanical properties of DN hydrogel samples.

We measure the dependence of bio-ink viscosity on shear rate $\dot{\gamma}$ in the range of 0.1 to 1000 1/sec. For all bio-inks, the viscosity is nearly constant at shear rates <1 1/sec and dramatically decreases at shear rates >10 1/sec. We take the value at the lowest shear rate 0.1 1/sec as the viscosity of a bio-ink.

## Uniaxial tensile test

Uniaxial tensile tests are performed using a motorized tension test stand (MARK-10 ESM303) equipped with a digital force gauge (MARK-10 M5-05) and two clamps (Mark-10 G1062). We use the same equilibrated hydrogel film in DMEM for rheological measurements to prepare the sample for tensile tests. Specifically, after rheological measurements, we detach the film from the geometry of the rheometer and use a die to cut the film into a dog bone-shape with the center part 2 mm in width and 13 mm in length. The sample is stretched at a fixed strain rate of 0.02/s until it fractures. Throughout this process, stress is recorded at a rate of 3 data points per second.

## Evaluation of the mixing capability of the multi-channel print nozzle

We use a digital camera (Hayear, 16 MP) equipped with a long working distance lens to monitor the mixing process in real time. To visualize the mixing, we incorporate red and green mica microparticles (50 μm) with a concentration of 0.5% (w/v), respectively, into a pair of bio-inks made of $PAM_{10}Alg_2$. The two bio-inks are simultaneously extruded through the dual-inlet nozzle's respective inlets, each at a constant flow rate of 0.15 μL/s, matching the flow rate used in DASP printing. The progression of the mixing process is visually demonstrated by initially activating the red channel, followed by activation of the green channel, and culminating with the deactivation of the red channel.

## 3D printing of the double-network and alginate hydrogel

We use our extrusion nozzle equipped with a dual-inlet print nozzle to demonstrate the printability of DN bio-inks (Fig. 4c, d; Fig. 6e). The method for DASP printing is the same as that used in our previous publication[8]. For filament-based printing, the movement of the nozzle is programmed by the slicer software (Ultimaker Cura 4.6).

Once the printing process is completed, the printed structure is left in the supporting matrix for 2 min for $PAM_{10}ALg_2$ bio-ink and 10 min for HA/alginate bio-ink. This period facilitates the solidification of the PAM and HA networks. Then, we immerse the supporting matrix in a bath containing 20 mM calcium and 120 mM NaCl at 37 °C for 10 min. This process not only dissociates the gelatin-supporting matrix but also completely crosslinks the alginate hydrogel. Following the removal of the supporting matrix, the printed 3D structures are kept in saline (150 mM NaCl in DI water) supplemented with 2 mM $CaCl_2$ for 24 h and imaged using a camera (Canon EOS 7D) equipped with a long working distance objective.

## DASP printing scaffolds encapsulated with Beta-TC-6 cells

For cell encapsulation within $PAM_{10}Alg_2$, we use the extrusion module equipped with a triple-inlet print nozzle, in which one channel is loaded with cell suspension and the other two are loaded with a pair of bio-inks. To prepare the cell suspension, we culture Beta-TC-6 cells (ATCC) in multiple T75 flasks to obtain a relatively large number of cells required by the fabrication. The culture medium is DMEM (4.5 g/L D-Glucose) supplemented with 10% FBS, 1% Pen/Strep, 0.0005% 2-Mercaptoethanol and 20 mM HEPES. To harvest the cells, we rinse each flask using 1x PBS, apply 3 mL Trypsin-EDTA for 2 min to detach the cells from the substrate, and subsequently add 6 mL DMEM culture media to neutralize the Trypsin-EDTA. The cell pellet is collected by centrifuging the cell suspension at 600 g for 5 min. The pellet is then resuspended in DMEM (high glucose, without calcium) containing 15% (w/v) dextran. The final concentration of the cell suspension is 20 million/mL.

To prepare the bio-inks, we respectively dissolve TZ-PAM and NB-PAM in DI water with a concentration of 10% (w/v). Subsequently, PEG is introduced into the solution with a concentration of 15% (w/v). The osmotic pressure from the PEG polymers is approximately 440 kPa[49], which is comparable to the osmotic pressure generated by 90 mM NaCl. To achieve a neutral pH of the bio-inks, we add 1 M HEPES stock buffer to reach a final HEPES concentration of 40 mM. Importantly, this pH adjustment is accomplished with minimal impact on the overall volume of the bio-ink. Taken PEG and HEPES together, the osmotic pressure resulting from polymers in the bio-ink is about 540 kPa, which is close to the physiological osmotic pressure of 730 kPa caused by ions at 20 °C. Finally, we dissolve the same amount of alginate in each of the two solutions with a concentration of 2% (w/v).

We load the cell suspension into a 250-μL syringe and load the pair of bio-inks respectively into two 1-mL syringes. During printing, these three syringes are driven by the extrusion module at the same speed such that the flow ratio between cell suspension and the two bio-ink channels is 1:4:4. We DASP print 5×5×4 lattice scaffolds (Supplementary Fig. 14) encapsulated with Beta-TC-6 cells. The printed scaffolds are transferred to a 24-well plate with one scaffold in 1 mL Beta-TC-6 culture medium per well. To study the distribution of cell-mimicking microparticles with an average diameter of 20 micrometers in the extruded bio-ink (Fig. 5a, iii), we replace the Beta-TC-6 cells with polystyrene microparticles at the same concentration of 20 million/mL in suspension.

To characterize the porosity of the aqueous two-phase system, we label the dextran phase with fluorescence and capture the distribution of the dextran phase in 3D using fluorescence confocal microscopy. Specifically, we add a trace amount of FITC-labeled dextran (2000 KDa) to the dextran phase with a concentration of 0.2% (w/v). Subsequently, we use the extrusion nozzle to mix the dextran phase with the counterpart phase containing 15% (w/v) PEG and 2% (w/v) alginate. The mixture is crosslinked by 20 mM calcium ion right before the imaging.

For cell encapsulation within HA/alginate, we first prepare the bio-inks by respectively dissolving TZ-HA and NB-HA in DMEM cell culture media (without calcium) supplemented with 30 mM HEPES with a concentration of 2 and 4% (w/v). Additionally, we dissolve the same amount of alginate in each of the two solutions with a concentration of 2% (w/v). Next, we directly mix the Beta-TC-6 cell suspension with the TZ-HA phase and load the pair of bio-inks in two syringes; the final cell density in the TZ-HA phase is 5 million/mL. Finally, we use the dual-inlet nozzle (Supplementary Fig. 10b) to mix the pair of solutions on-demand, extrude the bio-ink, and print the 5×5×4 lattice scaffolds encapsulated with Beta-TC-6 cells, following the same printing process described in Fig. 4. The printed scaffolds are transferred to a 24-well plate with one scaffold in 1 mL Beta-TC-6 culture medium per well.

## Compression tests

To perform compression tests on the printed scaffolds, we use a rheometer (Anton Paar MCR 301) with a normal force resolution of 0.5 mN, sufficient for capturing the small deformation of the scaffolds. To ensure that the samples remain fully hydrated throughout the tests, we conduct the experiments within an aqueous environment. Specifically, the sample is situated in an open container filled with saline containing 2 mM $Ca^{2+}$ that mimics the DMEM composition. We place the sample, along with the container, onto the bottom geometry of the rheometer. Subsequently, the upper geometry is lowered into the saline to establish contact with the sample. Throughout the compression measurements, the upper plate is programmed with a pre-defined moving profile, which exerts cyclic and/or large one-way compression at a constant strain rate of 0.005/s. The parameters monitored during the test include the normal force, gap size, and time. Utilizing the pre-measured dimensions of the samples, we calculate stress and strain values.

To quantify the hysteresis of gyroid structures made of DN bio-ink (Fig. 4f), we introduce energy dissipation efficiency (DE), which is defined as the ratio between the integrated area in the hysteresis loop and that under the compression curve:

$$DE = \frac{|\int_{load}\sigma d\epsilon| - |\int_{unload}\sigma d\epsilon|}{|\int_{load}\sigma d\epsilon|} \qquad (15)$$

Here, $\epsilon$ is the compression strain and $\sigma$ is the stress.

## Cytocompatibility of DASP 2.0

We use Beta-TC-6 cells to perform live/dead assay to assess the cytocompatibility of DASP 2.0. To prepare the working solution, stock solutions of propidium iodide (750 μM in Dulbecco's phosphate-buffered saline (DPBS)) and fluorescein diacetate (1 mM in dimethyl sulfoxide (DMSO)) are added to Hanks' balanced salt solution (HBSS), yielding final concentrations of 3.75 μM and 0.2 μM, respectively. To stain both the naked Beta-TC-6 cells and the encapsulated cells, we replace the culture media with the working solution at a volume of 1 mL per well for the 24-well plate. Fluorescence images are captured using a confocal microscope (Leica SP8) equipped with a 10x dry objective with a numerical aperture of 0.3. The image acquisition is performed under multitrack mode, where red fluorescence from propidium iodide is excited by a HeNe laser (552 nm) and collected through a bandpass filter of 600/700 nm. Green fluorescence from fluorescein diacetate is excited by an Argon laser (488 nm) and collected through a bandpass filter of 500/540 nm.

To visualize cells encapsulated in hydrogel particles, we acquire a stack of images along the z-axis to capture almost all the cells. Subsequently, these images are combined to generate a z-projection image, as shown in Fig. 5b and Fig. 6f, ii; this method enables a comprehensive assessment of the fraction of living cells. The survival rate is calculated as the ratio between the number of green dots and the total number of green and red dots.

## Glucose-stimulated insulin secretion (GSIS) test

We follow our previous method[8] to perform GSIS test on naked Beta-TC-6 cells and 5×5×4 DASP scaffolds encapsulated with Beta-TC-6 cells. During GSIS test, DASP printed scaffolds are placed in the 24-well plate with one scaffold per well and naked Beta-TC-6 cells are seeded in the 24-well plate with $1.2 \times 10^5$ cells per well, such that the number of encapsulated and naked Beta-TC-6 cells in each well are equivalent. Before the test, the naked and encapsulated Beta-TC-6 cells undergo a pre-incubation stage in a basal-glucose solution consisting of Krebs-Ringer bicarbonate buffer (KRB) supplemented with 2 mM D-glucose. The pre-incubation involves two cycles, with each cycle separated by a 1-hour interval. This pre-incubation process not only completely suppresses the cell pathway for insulin release, but also removes the

residual insulin that may be trapped within the scaffold. Following the pre-incubation cycles, the washing solution is substituted with 1 mL of fresh basal-glucose solution, which is then incubated for an additional hour. Subsequently, the entire medium is collected and transferred to 1.5-mL centrifuge tubes. Then, we add 1 mL high-glucose solution (KRB supplemented with 25 mM D-glucose) to the well with scaffold inside and further incubate for 1 h, at which the medium is once again collected. All the collected solutions are centrifuged at 600 g for 3 min and an aliquot of 600 μL is extracted from the supernatant. This approach ensures the effective exclusion of any potential cell debris. The insulin concentration in the collected supernatant is measured through mice insulin ELISA kits (Mercodia, Fisher Scientific, Cat. No. 10124710) and the release index is calculated by normalizing the insulin concentration upon stimulation to the basal value.

## Animal studies

Animal studies are performed following a protocol approved by the University of Virginia Animal Care and Use Committee (Protocol No. 4360-08-21). Animals are housed in cages supplied with food pile, water, and corncob bedding. The cages are located in an animal facility with a 12-h light/12-h dark cycle, a temperature of 22 °c, and a relative humidity of 50%. In our animal studies based on $PAM_{10}Alg_2$ (Fig. 5e–h), we use two types of DASP printed 5×5×4 lattice as scaffolds for transplantation. The first type is DN scaffolds the same as those tested for cytocompatibility and GSIS but without cells encapsulated. The second type is pure alginate single-network scaffolds printed by DASP 1.0. For each type, we transplant two scaffolds to the abdominal cavity of one immunocompetent C57BL/6 mouse (male, 12 weeks old, Jackson Laboratory) and use 6 mice per group. For the group transplanted with pure alginate single-network scaffolds, we sacrifice all 6 mice 7 days after the transplantation. The scaffolds fracture into pieces can barely be retrieved. For the group transplanted with DN scaffolds, we sacrifice 2 mice and retrieve the scaffolds, respectively, at 30, 60, and 120 days after the transplantation. The retrieved scaffolds are preserved in saline supplemented with 0.02% sodium azide. Subsequently, the scaffolds are imaged using a camera (Canon EOS 7D) and a digital microscope (Hayear, 16 MP), followed by compression tests.

In animal studies based on HA/alginate (Fig. 6g), we transplant two scaffolds made of DN HA/alginate hydrogel to the abdominal cavity of one immunocompetent C57BL/6 mouse (male, 12 weeks old, Jackson Laboratory) and use 3 mice in total. The scaffolds are retrieved 30 days after the transplantation.

## Statistical analysis and reproducibility

Data was analyzed using MATLAB for curve fitting and FIJI for image analysis. The data for cytocompatibility and GSIS test are shown as mean ± standard deviation (S.D.) with sample size $n = 5$. The representative image in Fig. 5a, iii is shown based on similar results observed in three repeated experiments. The representative photograph in Fig. 5g, iv is shown based on similar results observed for all retrieved scaffolds.

## Reporting summary

Further information on research design is available in the Nature Portfolio Reporting Summary linked to this article.

## Data availability

All data supporting the findings of this study are included in the manuscript and Supplementary Information, which are also available from the authors on request. Source data are provided with this paper.

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

## Acknowledgements

The authors thank Avery Baker and Gabriella Recce for assistance with initial chemical synthesis, and Dr. Jose Oberholzer for helpful discussions. L.H.C. acknowledges the support from NSF CAREER DMR-1944625, NSF CBET-2306012, the UVA LaunchPad for Diabetes, the UVA Coulter Center for Translational Research, Juvenile Diabetes Research Foundation (JDRF 1-INO-2022-1114-A-N), grant funding from Virginia's Commonwealth Health Research Board, and the UVA Center for Advanced Biomanufacturing.

## Author contributions

L.H.C. conceived the research. L.H.C. and J.Z. designed the research. J.Z. performed the research including hardware setup, chemical synthesis, physical and biological characterization, and data analysis. Y.W. and J.Z. performed the animal surgery. J.Z. and L.H.C. wrote the paper. Y.H. helped with cell culture, biological characterization, and animal surgery. All authors reviewed and commented on the paper.

## Competing interests

L.H.C. and J.Z. have filed a provisional patent (No. 63/538,260) based on the printing technique, biomaterials, and biomedical applications of DASP 2.0, and declare no other competing interest. Patent assigner: L.H.C. and J.Z.; assignee: University of Virginia; status: application submitted. The remaining authors declare no competing interests.
