## [Peer Review File · Nature Communications]

Voxelated bioprinting of modular double-network bio-ink dropletsEditorial Note: Parts of this Peer Review File have been redacted as indicated to remove third-party material where no permission to publish could be obtained.

REVIEWER COMMENTS

Reviewer #1 (Remarks to the Author):

The authors present a voxelated bioprinting method that enables assembly of interpenetrating alginate and polyacrylamide (PAM) double network hydrogel droplets. High MW PAM polymers were functionalized with either tetrazine or norbornene, which upon mixing formed a network via additive-free click reaction. This modification represents some advantages over conventional radical-mediated polymerization of acrylamide monomers, most notably in tunable mechanical properties.

The study builds on a previously established method that features the digital assembly of spherical particles (DASP). These highly viscoelastic aqueous droplets are first deposited in a yield-stress fluid and subsequently assembled into 3D structures by controlled polymer swelling. Moreover, the present work develops a theoretical framework to describe crosslinking kinetics and stiffness of hydrogels. Lastly, the study showcases the printing of complex architectures and characterizes their cytocompatibility in vitro.

Detailed and accurate comparisons of experiment and theory are a strength of this manuscript, as well as the increased mechanical tunability of the printed structures. The cytocompatibility of PAM (when compared with the authors' previous work on the DASP1.0 alginate only system) represents a significant weakness, as noted by the authors.

I suggest publication after considering the following comments.

1. Line 115/135-136: I recommend replacing "Crosslinker" with "Crosslink", as a crosslinker can be reacted or unreacted, while "crosslink"³ is more clearly defined as a reacted crosslinker serving as a physical crosslink.
2. Does the theoretical prediction in Fig 2e (dashed line) include the prefactor? If not, since prefactors, such as the Flory's ratio have been considered, it is good to compare the experimental results with the full prediction (rather than just the scaling) and discuss where the errors may arise. In addition, only one value of f_x is used among all the sample compositions. I recommend testing more f_x to justify the 9/5 scaling factor.
3. The authors claimed that the total shear modulus of the DN hydrogel is the sum of the two components (line 214), and the modulus-critical strain prediction (Eq. 14) is purely derived from the Kuhn step of PAM network (from Eq. 10). Therefore, it is more reasonable to subtract the modulus contribution from Alg when comparing the ratio between the two moduli (Line 269). My calculation gives $((7-4.5)/(20-4.5))^{(-2/9)} \sim 1.5$ instead of 1.3.
4. The theoretical framework laid out describes crosslinking kinetics and stiffness, and presents a diagram-of state to delineate the mechanical properties. However, subsequent experiments are conducted on DN systems. It would be worthwhile to embed the presented theory into a broader theoretical framework that also takes account of the DN contribution. In this regard, other studies

might be relevant (Khiêm et al., *Macromolecules*, 2019; Wang & Wei, *Soft Matter*, 2011; Gong et al. *Advanced Materials*, 2003).

5. Figure 3d highlights the stress-strain behavior of PAM/Alg hydrogels. Further extensibility/tensional studies should be carried out on DASP printed constructs rather than on bulk hydrogels to characterize the connectivity (scaffold strength) between printed droplets following swelling.

6. Can a statement be made about achievable resolution? Is there a practical lower limit for the diameter of the droplets that still enables high cell viability given increasing shear forces during printing?

7. How does the swelling influence printing fidelity (model vs actual print)? Is swelling in droplets isotropic? Please specify a swelling ratio for the chosen material compositions. What solutions do the authors propose for materials with low to no swelling ratio to fuse individual droplets (e.g. printing droplets in a slightly overlapping fashion)?

8. Given the material composition and lack of proper adhesion sites, it is unlikely that a broad range of cells (eg. mesenchymal cells) can be sustained by the scaffold material (either in vitro or in vivo). Can the authors comment on broader cell compatibility and what kind of modifications can be done to allow for this?

9. Relatively free migration of nutrients in areas with larger pores (in-between droplets) stands opposed to the microporosity within droplets. Fig 5a,ii-iii depicts the two-phase system resulting following phase separation of the DN-ink/dextran bioink but a more thorough characterization of the resulting porosity is missing. To judge whether this system actually benefits cells, a more in-depth discussion and characterization is required (e.g. pore size distribution, pore inter-connectivity, differences in diffusion between bulk hydrogel and phase-separated hydrogel, necessity of enzymatic breakdown of scaffold for cell migration etc.). Furthermore, how does this multi-scale porosity translate into different cell responses?

10. It is stated that PAM polymers are not optimal regarding their long-term cytocompatibility. Yet, the bulk of the characterization and discussion is based on using functionalized PAM polymers. The authors suggest other materials (HA, PEG) that could be used instead. A brief discussion of what constitutes material compatibility with DASP protocols should be added.

11. Regarding viability of cells: how does the surface compare to the core of droplets? As the method relies on the encapsulation within a more dense bulk material (with microporosity), viability might be dependent on the diffusion distance. A characterization of cell viability vs. localization within droplets (and differences in viability/metabolic activity) is important. The authors should also comment on physical limitations (diffusion limits, nutrient availability) in single droplets vs. aggregate structures. Herein, it would be helpful to quantify the viability limit from the outside of the droplets.

12. Line 528 should read: "more cytocompatible" as there is still relatively low viability exhibited in the crosslinked DN.

13. The authors state that the scaffold can be retrieved from the implantation site of immunocompetent mice after 4 months without significant impairments to the structural integrity. A photograph of the construct before and after is shown. However, the reviewer is missing any kind of histological assessment showing (immune) cell infiltration, blood vessel infiltration, matrix deposition etc. As no significant changes in mechanical characteristics (stress-strain) were observed following the in vivo explantation (Fig 5h), the scaffold materials seem rather inert to in

vivo environments.

Minor comments:

- Fig 3e: There are two “PAM10Alg2”
- Line 121: “[...] decreases with rapidly with [...]” should be “decreases rapidly with”
- Line 263: 2.5% TZ and 0.5% NB should give you NB/TZ=0.2, instead of 0.2 in Fig. 2a
- Line 419: Release index = low/high or high/low glucose media?
- Line 532: “[...] relies on-demand [...]”; missing “on” after relies
- Line 539: “[...] enabling the fabrication highly complex [...]”; missing “of” after fabrication

Reviewer #2 (Remarks to the Author):

The authors have introduced the novel concept of Digital Assembly of Spherical Particles (DASP) as a means of generating larger volumes of engineered tissue constructs through the utilization of voxelated bioprinting technology. This particular methodology has undergone comprehensive development and establishment by the authors, culminating in the publication of multiple articles in various academic journals. The principal innovation of this study lies in the application of a new ink material aimed at yielding more flexible structures as opposed to the single-network Polyacrylamide (PAM) hydrogel. The objective is to match the mechanical properties of the engineered tissue constructs with those of native tissues and organs. Nevertheless, a clear rationale for the necessity of elasticity in the bioink material remains elusive.

The authors asserted that the DASP hydrogel is created by crosslinking PAM polymers via a biologically compatible, bioorthogonal chemistry. However, it is noteworthy that the DASP hydrogel does not directly encapsulate cells, which raises questions regarding the unequivocal characterization of its "biocompatibility." Furthermore, the definition of bioorthogonal chemistry implies a direct interaction between specific cellular sites and the material, a criterion not met by the DASP hydrogel, potentially constituting an overstatement.

Despite the authors' comprehensive elucidation of the DASP hydrogel within the context of 3D printing, its attention from the bioprinting research field is hindered by its limited viability. Even with the addition of a cell-laden layer, the system failed to meet the expected viability standards post-printing. Additionally, the inherent properties of the applied materials precluded the anticipation of significant biological functionality.

In light of these considerations, this manuscript must undergo substantial refinement to enhance its originality and to furnish concrete scientific findings pertaining to the bioink's suitability for utilization in bioprinting before its consideration for publication.

Reviewer #3 (Remarks to the Author):

1. This manuscript discusses progress in the development of biomaterial ink, the technology used for printing it, and the possible applications of the material in the form of 3D objects. The author highlights advancements in printing voxels based on alginate and PAM, representing an improvement over the previously developed DASP1, which could only print alginate. However, DASP1 faced toxicity issues due to unreacted polyacrylamide. The authors also present in vivo data assessing the safety of the 3D printed materials. The article provides interesting information and adds to the body of the literature but shall require major revision before reconsideration.

2. The authors are encouraged to provide a more thorough discussion of their previous development, DASP1, as its significance remains unclear to the reviewer. The study's importance should extend beyond the comparison of PAM/alginate versus alginate and the hardware development for printing the mixture. A more comprehensive discussion in this regard would enhance the overall clarity and impact of the study.

3. The reported method appears primarily suited for generating small size constructs. Has the author explored the size variability associated with this method? It would be valuable to investigate the range of sizes feasible through this approach, specifically identifying the maximum and minimum constructs that can be 3D printed and shall be compared with their DASP1. Additionally, it is essential to elucidate the material dependency of this approach concerning the dimensions of the constructs.

4. The details regarding the mechanical properties of the constructs, particularly in terms of compression, need to be more thoroughly elaborated with respect to the size of the scaffolds. It is crucial to investigate whether the voxel crosslinking is linearly influenced by the size of the scaffold, potentially leading to variations in compression modulus. Additionally, exploring how compression force may impact the interaction of voxels becomes pertinent. A more in-depth analysis of these aspects may improve the understanding of the interplay between scaffold size, voxel crosslinking, and compression behavior, providing valuable insights into the mechanical characteristics of the constructs.

5. While the printing of voxels introduces multiscale porosity, potentially facilitating nutrient diffusion into the scaffold, I couldn't see the data related to construct porosity. It would be valuable to include information on the porosity of the construct, particularly examining both macro and micro porosity. Utilizing uCT and SEM may be helpful in this regard. This additional information is crucial for a comprehensive understanding of the scaffold's porosity characteristics, and its suitability for nutrient diffusion and overall performance.

6. The animal study is insufficient to establish the material's usefulness. While it demonstrates the lack of toxicity, the significance of this approach should extend beyond a mere safety test. Notably, there is an absence of data on whether the porosity contributes to vascularization within the

construct?. To fully show the application and performance of the reported material/method in vivo, it is crucial to include comprehensive data addressing aspects such as vascularization. This will strengthen the study's overall impact.

7. In line with previous comment what was/is the fate of the material that implanted in vivo? It was mentioned that the implant was retrieved after 7 days, first the 7 day window may be too short for the analysis. Second I don't see the histology data, or any immunohistochemistry information. Do we know if the porous structure remained permeable during the study? What are the swelling properties of the voxels? There are many data missing both for in vivo studies and also concerning the fate of the material in vitro (swelling, degradation plus the mechanical studies of the material behaviour in vitro e.g. in SBF) and in vivo. If degraded then the rate and kinetic of degradation need to be provided.

8. The rationale behind employing on-demand mixing of the TZ-PAM and NB-PAM polymers for printing the DN bio-300 ink is not clearly defined. The authors should first show the necessity of this specific design and subsequently address its applicability to other bioinks. It is crucial to articulate the importance of undertaking such actions in comparison to their previously developed alginate-only method. Doing so may enhance the reader's understanding of the motivation behind the chosen approach and its potential advantages over alternative methods, particularly in relation to their prior work with alginate-only materials.

9. In the introduction section, the manuscript did not provide a clear and comprehensive literature review on the current state-of-the-art. While the importance of voxel printing was compared with droplet printing and the importance of printing viscoelastic ink in this approach but the authors could also indicate the microgel printing approach or printing the emulsion based biomaterial ink that also can result in constructs with good mechanical properties and may require a simpler approach compared to printing voxels. In general providing advantages and disadvantages of different biomaterials and printing methods, and the gaps and challenges that the proposed biomaterial ink aims to overcome.

10. The authors encapsulated human islets in individual particles and demonstrated that the scaffolds allow for responsive insulin release to highlight the potential of DASP in biomedical applications, nevertheless there is a wide range of studies on encapsulating human islets in gel beads including alginate based materials developed by simple bead formation, microfluidic beads and so on with a diverse range of control on the porosity of the beads to induce release of insulin while preventing the damage to the encapsulated islets. While there is no information here on the nutrient and oxygen diffusion to the scaffolds not the role of the scaffold in protecting the cells from immune system, nor also on cellular overgrowth, then the authors need to carefully revise, adding limitations to their studies and provide a better discussion compared to the existing state of the art. Then the authors need to better discuss the potential application of the DASP realistically.

11. The authors also should better define and justify the ratio of TZ-PAM and the NB-PAM and alginate and its connection to the printability of the biomaterial ink. Also the crosslinking density of the material is not clear. Has there been an optimisation study?

12. In the conclusion section, the authors should also acknowledge or address some potential issues or challenges of the study, such as the reproducibility and scalability of the printing method, the biocompatibility and immunogenicity of the biomaterial ink and, the long-term stability and functionality of the constructs, and the clinical relevance and applicability of the biomaterial ink for biomedical engineering applications. Therefore, I recommend that the paper be revised to address these limitations and challenges before it can be accepted for publication.

Reply to Reviewers' Comments

We thank the reviewers for constructive comments, which help us improve the quality and clarity of the manuscript.

Reviewers' comments are in **Times New Roman (black)**; our responses are in **Arial (blue)**. We include revisions to the text for convenience in **green**, and initial version in **orange**.

Reviewer #1

The authors present a voxelated bioprinting method that enables assembly of interpenetrating alginate and polyacrylamide (PAM) double network hydrogel droplets. High MW PAM polymers were functionalized with either tetrazine or norbornene, which upon mixing formed a network via additive-free click reaction. This modification represents some advantages over conventional radical-mediated polymerization of acrylamide monomers, most notably in tunable mechanical properties.

The study builds on a previously established method that features the digital assembly of spherical particles (DASP). These highly viscoelastic aqueous droplets are first deposited in a yield-stress fluid and subsequently assembled into 3D structures by controlled polymer swelling. Moreover, the present work develops a theoretical framework to describe crosslinking kinetics and stiffness of hydrogels. Lastly, the study showcases the printing of complex architectures and characterizes their cytocompatibility *in vitro*.

Detailed and accurate comparisons of experiment and theory are a strength of this manuscript, as well as the increased mechanical tunability of the printed structures. The cytocompatibility of PAM (when compared with the authors' previous work on the DASP1.0 alginate only system) represents a significant weakness, as noted by the authors.

We thank the reviewer for complementary comments about our work. We would like to add that compared with *in situ* free radical polymerization of acrylamide monomers, which is cytotoxic and would kill all cells, our strategy is much more cytocompatible in the context of cell viability. Yet, we agree with the reviewer that compared to DASP1.0, which printing pure alginate only, the cytocompatibility of the double-network bio-ink reported in this work is lower in the context of cell viability.

The relative lower cytocompatibility of DASP 2.0 is attributed to the chemical nature of polyacrylamide (PAM), which we discovered to be slightly cytotoxic as free polymers in solution (**Supplementary Fig. 15** in the revised submission) but becomes cytocompatible when crosslinked. The cytocompatibility of the double-network bio-ink can be improved by replacing the PAM backbone with different polymer species such as polysaccharide, as commented in the **Conclusion** section in the original submission.

To address the reviewer's concern, in the revised manuscript, we have synthesized another double-network bio-ink based on hyaluronic acid, which possesses significantly improved cytocompatibility with 81% cell viability, which represents the upper limit compared to that for existing extrusion-based bioprinting technologies¹. More details can be found in the reply to the reviewer's specific comments below.

I suggest publication after considering the following comments.

1. Line 115/135-136: I recommend replacing “Crosslinker” with “Crosslink”, as a crosslinker can be reacted or unreacted, while “crosslink” is more clearly defined as a reacted crosslinker serving as a physical crosslink.

We have replaced “crosslinker” by “crosslink” throughout the main text.

2. Does the theoretical prediction in Fig 2e (dashed line) include the prefactor? If not, since prefactors, such as the Flory’s ratio have been considered, it is good to compare the experimental results with the full prediction (rather than just the scaling) and discuss where the errors may arise. In addition, only one value of f_x is used among all the sample compositions. I recommend testing more f_x to justify the 9/5 scaling factor.

The dashed line in **Fig. 2e** is the prediction of our scaling theory [eq. (9)], which does not include prefactors. For scaling theory, the values of prefactors are typically on the order of unity and should be determined by comparing with experiments. For instance, our theory predicts that the saturated shear modulus of the hydrogel is $k_B T$ per volume pervaded by the polymer section between two neighboring functional sites, $G_{max} \approx \frac{k_B T}{d_x^3}$ [eq. (6)]. The size d_x of the polymer section is given by eq. (7), $d_x \approx b \left(\frac{f_x^{-1}}{n_K} \right)^{\frac{3}{5}} \approx 1.6 \text{ nm} \left(\frac{40}{6.5} \right)^{\frac{3}{5}} \approx 4.8 \text{ nm}$; this gives $G_{max} \approx 37 \text{ kPa}$. Comparing this value against the experimentally measured value 22 kPa, one obtains the value of the prefactor 0.6.

We appreciate the reviewer’s comment on testing more f_x to further strengthen the theory. Yet, we notice that to test a scaling theory, it is important to cover different scaling regimes, which often requires varying the governing parameter in a wide range, such that the scaling exponent can be determined. This is exactly what has been done in the original submission: we vary the polymer concentration in wide range to determine the scaling exponent. In addition, the value of f_x explored in this study was chosen based on the theory, which predicts that the max stiffness the hydrogels can reach is $\sim 20 \text{ kPa}$, which is sufficient for most biomedical applications. Further, this work focuses on double-network hydrogels suitable for DASP printing rather than theory. Thus, we opt to explore other f_x values in future studies dedicated to theoretical understanding of hydrogel mechanics.

To address this review comment, we have added discussions on prefactors for scaling theory in the main text on page 11.

“Note that scaling theory does not include prefactors, which typically have values on the order of unit. For instance, our theory predicts that the maximum hydrogel stiffness is $G_{max} \approx 37 \text{ kPa}$ [eq. (8)]. Comparing the prediction against the experimentally measured value 22 kPa, one obtains a prefactor of 0.6.”

3. The authors claimed that the total shear modulus of the DN hydrogel is the sum of the two components (line 214), and the modulus-critical strain prediction (Eq. 14) is purely derived from the Kuhn step of PAM network (from Eq. 10). Therefore, it is more reasonable to subtract the modulus contribution from Alg when comparing the ratio between the two moduli (Line 269). My calculation gives $((7-4.5)/(20-4.5))^{(-2/9)} \sim 1.5$ instead of 1.3.

That is an excellent comment and thanks for catching this. Additionally, given that the extensibility is measured after the hydrogels reaching equilibrium, we notice that it is more reasonable to use the shear modulus of hydrogels at equilibrium in DMEM in the calculation. The shear moduli after

reaching equilibrium in DMEM of $\text{PAM}_{10}^{1/4}\text{Alg}_2$, $\text{PAM}_{10}^{1/1}\text{Alg}_2$, and Alg_2 are 7.4 (Fig. 3e, Fig. R1), 11.5 (Fig. 3b, e), and 4.5 kPa (Fig. 3b, e), respectively. Therefore, $\lambda_{max,1/4}/\lambda_{max,1/1} \approx [(G_{1/4} - G_{alg})/(G_{1/1} - G_{alg})]^{-2/9} \approx 1.2$.

Correspondingly, we have revised the main text “This formulation results in a lower network shear modulus $G \approx 6.4$ kPa (Fig. 2a) ...” to

“This formulation results in a lower network shear modulus of DN hydrogel $G \approx 7.4$ kPa (Supplementary Fig. 9)...”

“Considering that the stiffness of the DN network with TZ/NB ratio 1/1 is $G_{1/1} \approx 20$ kPa, the ratio of the extensibility of the softer DN hydrogel to that of the stiffer one is predicted to be $\lambda_{max,1/5}/\lambda_{max,1/1} \approx (G_{1/5}/G_{1/1})^{-2/9} \approx 1.3$.” to

“Considering that the ionic crosslinked alginate network has negligible contribution to λ_{max}^T and the equilibrium shear modulus $G_{1/4}$, $G_{1/1}$, and G_{alg} are respectively 7.4, 11.5, and 4.5 kPa (Figs. 3b, e), the ratio of the extensibility of the softer DN hydrogel with NB/TZ ratio 1/4 to that of the stiffer one with NB/TZ ratio 1/1 is predicted to be $\lambda_{max,1/4}^T/\lambda_{max,1/1}^T \approx [(G_{1/4} - G_{alg})/(G_{1/1} - G_{alg})]^{-2/9} \approx 1.2$.”

In addition, we add the result of oscillatory frequency sweep on $\text{PAM}_{10}^{1/4}\text{Alg}_2$ gel as Supplementary Fig. 9 in the revised Supplementary Information.

Supplementary Fig. 9. Dynamic mechanical properties of a double-network hydrogel. Dependencies of storage (G' , solid line) and loss (G'' , dashed line) moduli of completely crosslinked double-network $\text{PAM}_{10}^{1/4}\text{Alg}_2$ with mismatched TZ and NB grafting ratios. $\text{PAM}_{10}^{1/4}\text{Alg}_2$ is made from TZ-PAM with a grafting ratio of 2.53% and NB-PAM with a grafting ratio of 0.63%. The composition of a crosslinked hydrogel is denoted as $\text{PAM}_x^z\text{Alg}_y$, where x is the concentration in (w/v)% for PAM consisting of equal amount of TZ-PAM and NB-PAM, y is the concentration of alginate, and z is ratio between NB and TZ grafting ratios.

4. The theoretical framework laid out describes crosslinking kinetics and stiffness, and presents a diagram-of state to delineate the mechanical properties. However, subsequent experiments are conducted on DN systems. It would be worthwhile to embed the presented theory into a broader theoretical framework that also takes account of the DN contribution. In this regard, other studies might be relevant (Khiêm et al., Macromolecules, 2019; Wang & Wei, Soft Matter, 2011; Gong et al. Advanced Materials, 2003).

We appreciate the reviewer's comment on embedding the presented theory into existing theoretical framework for DN hydrogels. However, doing so is very challenging because of limited knowledge in network topology, as discussed in the last paragraph of Section "**Stiffness and extensibility of DN hydrogels**".

This work focuses on the development of a DN bio-ink suitable for DASP printing, which requires a well-controlled crosslinking kinetics of the bio-ink. The behavior of alginate has been well characterized in our previous studies². The most important question remaining is to understand the behavior of the PAM single-network hydrogel. Thus, we decided to elucidate the crosslinking kinetics, stiffness, and extensibility of PAM single-network hydrogels. This knowledge successfully guided us to develop PAM/Alg DN bio-ink for DASP printing.

The two papers suggested by the reviewer (Khiêm et al., *Macromolecules*, 2019; Wang & Wei, *Soft Matter*, 2011), respectively, presented elegant micromechanical and continuum mechanical models that offer insights into how the second weak network help a DN hydrogel reach the theoretical elastic extensibility. In the revised manuscript, we have added these two references to our discussion on the difference between our DN hydrogels and the classic DN hydrogels.

5. Figure 3d highlights the stress-strain behavior of PAM/Alg hydrogels. Further extensibility/tensional studies should be carried out on DASP printed constructs rather than on bulk hydrogels to characterize the connectivity (scaffold strength) between printed droplets following swelling.

That is an excellent point. To test the connectivity between printed droplets, we DASP printed a 1D filament and conducted a tensile test at 0.05/sec. The 1D filament has a tensile breaking strain ϵ_f of 0.7, which is smaller than that of bulk hydrogel of 1.0. The lower breaking strain may be caused by the weakened droplet connectivity. Nevertheless, the 1D filament shows a much higher breaking strain compared with PAM and Alg single-network hydrogels. In the revised manuscript, we have added a paragraph after the discussion of Fig. 4c in **DASP printing of DN hydrogels**:

"In a DASP printed structure, the bridges between neighboring droplets may be weaker than the bulk hydrogel, which would impair the mechanical robustness of the structure. To this end, we quantify the extensibility of a DASP printed 1D filament that comprises a sequence of interconnected droplets (**Supplementary Fig. 12**). As expected, the 1D filament has a tensile breaking strain ϵ_f of 0.7, which is slightly lower than that of bulk hydrogel of 1.0 (**Figs. 3c, d**). Nevertheless, the breaking strain of the 3D filament remains much larger than the constituent single-network hydrogels ($\epsilon_f \approx 0.38$). These results further support that DASP enables the fabrication of integrated structures consisting of interconnected droplets."

Supplementary Fig. 12. Snapshots of a DASP printed 1D filament under uniaxial elongation. The bio-ink is PAM₁₀Alg₂ and the tensile strain rate is 0.05/sec. The left, middle, and right panels are, respectively, captured at the beginning of the tensile test, right before breaking, and right after breaking. Scale bar, 1mm

6. Can a statement be made about achievable resolution? [continued]

The printing resolution is determined by the smallest size of a single droplet that can be printed with good fidelity. For our current system, the lower limit is about 300 μm , as shown in **Fig. R1** (**Fig.6a** in our previous work ²).

Fig R1. Dependence of hydrogel particle diameter on various combinations of injection volume and speed. Dashed line: predicted particle diameter d_T based on the injection volume V : $d_T = (6V/\pi)^{1/3}$. Solid line is the best weighted fit to the experimental data: $d_E = (1.12 \pm 0.04) d_T$, suggesting the particles swell by 12%. Inset: Hydrogel particles swell by nearly 11% in diameter at 10 min. A particle reaches 80% of equilibrium swelling at about 200 s. Adapted from Ref. ²

Is there a practical lower limit for the diameter of the droplets that still enables high cell viability given increasing shear forces during printing?

The cell viability is not limited by the droplet diameter if it is much larger than the size of single cells.

In DASP, printing a droplet consists of three stages: bio-ink extrusion, nozzle detachment from the droplet, and droplet relaxation, as detailed in our previous study³. The shear rate associated with extrusion is relatively low and is independent of droplet diameter. By contrast, the shear rate associated with nozzle detachment is relatively high, as it involves detaching the nozzle from the droplet at a relatively high acceleration, not speed, and that it increases with the decrease of droplet diameter. Thus, we assume that the reviewer is asking whether the shear force during detachment stage can impair cell viability, especially when the droplets become small.

Based our previous studies², the cell damage due to the shear stress is negligible because the shear occurs only at the surface of the droplet and does not directly apply to cells, which are encapsulated and protected by the droplet. However, when the droplet becomes very small, say tens of micrometers, and are only slightly larger than size of single cells, shear force may induce cell damage. This is something worthy of explorations but is beyond the scope of this work.

7. How does the swelling influence printing fidelity (model vs actual print)? Is swelling in droplets isotropic? Please specify a swelling ratio for the chosen material compositions. [continued]

The droplet swelling has negligible impact on printing fidelity as the swelling has been accounted in the printing, where the extrusion volume of each droplet is intentionally chosen to be smaller than the desired final volume to compensate the effects of swelling.

According to our previous study, the swelling ratio for pure alginate bio-ink is 12% in diameter and the swelling is isotropic. However, we appreciate the reviewer's comment that the swelling ratio may be different for the DN bio-ink. Thus, we have performed additional experiments to show that the swelling of DN droplets is isotropic (shown below and **Supplementary Fig. 11** in revised Supplementary Information). For droplets made the PAM/Alg DN bio-ink, the swelling ratio is 14%.

Accordingly, we have revised the main text

“Building on our previously established knowledge and leveraging the newly developed extrusion module, we successfully create a hollow sphere consisting of only one layer of interconnected yet distinguishable DN hydrogel particles.”

to

“Building on our previously established knowledge and leveraging the newly developed extrusion module, the printed droplets in the supporting matrix undergo isotropic swelling by 14% in diameter (**Supplementary Fig. 11**), enabling them to coalesce into a hollow sphere consisting of only one layer of interconnected yet distinguishable DN hydrogel particles.”

Supplementary Fig. 11. Swelling of DASP printed DN bio-ink droplets. Photos of DASP printed DN bio-ink droplets in the supporting matrix until equilibrium. The droplets are made of DN PAM₁₀Alg₂ hydrogel. Scale bar, 500 μ m.

What solutions do the authors propose for materials with low to no swelling ratio to fuse individual droplets (e.g. printing droplets in a slightly overlapping fashion)?

When being printed, the bio-ink is not a solid network but a polymer solution, where polymer chains overlap and repel with each other, resulting in an osmotic pressure higher than that of the supporting matrix. The difference in osmotic pressure drives water influx into the droplet. As a result, the bio-ink droplet must swell in the supporting matrix.

The extent of swelling before being crosslinked is controlled through swelling kinetics. Yet, as the reviewer pointed out, there might be cases where the swelling ratio is low, and one can adjust the distance between two neighboring droplets to ensure individual droplets partially coalesce before completely crosslinked.

8. Given the material composition and lack of proper adhesion sites, it is unlikely that a broad range of cells (eg. mesenchymal cells) can be sustained by the scaffold material (either in vitro or in vivo). Can the authors comment on broader cell compatibility and what kind of modifications can be done to allow for this?

We thank the reviewer for this excellent comment. The current DN bio-ink presents a cornerstone for our ongoing efforts in the development of modular biomaterials for understanding and controlling cell-matrix interactions. Using click chemistry, extracellular matrix (ECM) derived peptides such as RGD groups can be conjugated to the polymers for cell engagement. Using degradable crosslinkers such as matrix metalloproteinase (MMP)-degradable peptides would make the hydrogels suitable for 3D culture. We have started these chemical modifications and hope to present the results to the community soon.

In the context of cell compatibility, as replied to this reviewer's general comment, in the revised manuscript we have added an additional experiment, in which we replace the PAM-based polymer by hyaluronic acid (HA)-based polymer, featuring a cytocompatible polymer backbone and much more redundant free NB groups for further chemical modifications and crosslinking. More details about the TZ-HA/NB-HA hydrogels can be found in the reply to **Comment #10**.

9. Relatively free migration of nutrients in areas with larger pores (in-between droplets) stands opposed to the microporosity within droplets. [continued]

As the reviewer pointed out, a DASP printed scaffold has multiscale porosity: (1) micrometer-scale pores determined by interstitial space among interconnected droplets, and (2) nanoscale pores determined by the hydrogel network mesh size. These two kinds of pores complement with each other in enabling efficient nutrient transport. The large pores allow free transport of nutrients, whereas the small pores are large enough for diffusion-based nutrient transport through individual droplets. Such a multiscale nutrient transport ensures the viability of encapsulated cells, as detailed in our previous work².

Fig 5a,ii-iii depicts the two-phase system resulting following phase separation of the DN-ink/dextran bioink but a more thorough characterization of the resulting porosity is missing. To judge whether this system actually benefits cells, a more in-depth discussion and characterization is required (e.g. pore size distribution, pore inter-connectivity, differences in diffusion between bulk hydrogel and phase-separated hydrogel, necessity of enzymatic breakdown of scaffold for cell migration etc.). [continued]

Compared with mixing cells with the bio-ink directly, the two-phase system significantly improves cell viability, as shown by the contrasting result between the cell viability in **Fig. 5c** and **Fig. S8** in original submission (**Fig. S15** in revised submission). Potential benefits pointed out by the reviewer, such as differences in diffusion and necessity of enzymatic degradation, are useful for subsequent biomedical applications but are beyond the scope of this work, which focuses on voxelated bioprinting of DN bio-inks.

However, we do appreciate the reviewer's comment on in-depth characterization of networks formed by the two-phase system. In the revised manuscript, we have performed experiments to further characterize the porosity of the two-phase system in 3D. Specifically, we use fluorescent dextran to label the cell phase and obtain the 3D image the hydrogel using confocal microscopy. Within the crosslinked hydrogels, interconnected branches prevail and island-like cell phases with an average size of 10 micrometers are dispersed throughout the whole material, as shown below or by **Supplementary Fig. 13** in the revised manuscript.

Supplementary Fig. 13. Microstructure of hydrogels formed by aqueous-two phase system. Fluorescence confocal microscopy images depicting the PEG/dextran aqueous two-phase system mixed by the extrusion nozzle. Green phase, dextran with a concentration of 15% (w/v) and FITC labeled dextran with a concentration of 0.2% (w/v). Dark phase: PEG with a concentration of 15% (w/v) and alginate with a concentration of 2% (w/v), which is crosslinked by 20 mM calcium ion. The images are generated via 3D reconstruction from multiple 2D images.

Furthermore, how does this multi-scale porosity translate into different cell responses?

We assume the reviewer is asking how the multi-scale porosity affects cell behavior such as migration, differentiation, etc. We do not know exactly the answers at this point. These are interesting topics would be worth to explore in future studies.

10. It is stated that PAM polymers are not optimal regarding their long-term cytocompatibility. Yet, the bulk of the characterization and discussion is based on using functionalized PAM polymers. The authors suggest other materials (HA, PEG) that could be used instead. A brief discussion of what constitutes material compatibility with DASP protocols should be added.

The combination of polyacrylamide (PAM) and the alginate (Alg) represents a proof-of-concept DN bio-ink for voxelated bioprinting. Although the completely crosslinked PAM is cytocompatible, we discovered that when a solution of free PAM polymers is premixed with cells, it exhibits cytotoxicity (**Supplementary fig. 8** in the original submission) and poses challenges for cell encapsulation. To this end, we design a triple-inlet nozzle to mix cells with bio-ink and apply the two aqueous phase system to temporally separate the cells from the free PAM polymers before the hydrogel is fully crosslinked. Nevertheless, despite the cell viability remains 60%, it is much lower than that of DASP 1.0 using pure alginate bio-ink.

Indeed, all *three* reviewers are concerned about the relatively low cytocompatibility of the PAM/Alg DN bio-ink. As this reviewer noted, in the original submission, we suggested that this issue can be addressed by the replacing the PAM-based polymer backbone with other biopolymers such as hyaluronic acid.

In the revised manuscript, we have performed additional experiments to demonstrate our conjecture, in which we synthesize a new pair of TZ and NB functionalized polymer by replacing the PAM backbone with hyaluronic acid (HA). We perform a series of tests on the mechanical robustness, printability, cytocompatibility, and *in vivo* stability of HA/Alg DN hydrogel, as shown by **Fig. 6** in the revised manuscript.

The results and associated discussion on the HA/Alg DN bio-ink are included a new section in the main text, as shown below:

Fig. 6. Mechanical properties, printability, and biocompatibility of a DN hyaluronic acid/alginate (HA/Alg) bio-ink. (a) HA is functionalized with either norbornene (NB) or tetrazine (TZ) groups. Upon mixing, TZ-HA and NB-HA are crosslinked via the click reaction between TZ and NB groups, following the same crosslinking mechanism as that for creating a PAM network (Fig. 1b). The grafting ratio of TZ-HA and NB-HA are, respectively, fixed at 26.8% and 26.7%. (b) Dependencies of storage (G' , solid lines) and loss (G'' , dashed lines) moduli of completely crosslinked single-network HA and DN HA/alginate hydrogels on oscillatory shear frequency. Single-network hydrogel (HA): TZ-HA (2%, w/v) + NB-HA (4%, w/v). The DN hydrogel (HA/alginate) is prepared by mixing a pair of bio-inks, TZ-HA (2%, w/v) / alginate (2%, w/v) + NB-HA (4%, w/v) / alginate (2%, w/v). (c) Representative photos of the DN HA/alginate hydrogel under uniaxial tensile test at a fixed strain rate of 0.02/sec. The left, middle, and right panels, respectively, are captured at the beginning of the tensile test, right before fracture, and right after fracture. Samples are 13 mm in length and 2 mm in width of the center part. (d) Stress-strain behavior of the single-network HA and DN HA/alginate hydrogels in (b). (e) A DASP printed 5x5x4 lattice scaffold consisting of 100 interconnected yet distinguishable DN HA/alginate hydrogel particles. Upper panel: top view; lower panel: side view. Scale bar, 1 mm. (f) Cytocompatibility of the HA/alginate DN bio-ink. (i) A pair of HA/alginate bio-inks pre-loaded with cells are printed using a two-channel print nozzle. (ii) A representative fluorescence confocal microscopy image from live/dead assay of DASP encapsulated MIN6 cells within the DN HA/alginate hydrogel. Scale bar, 100 μ m. (iii) Viability of naked and DASP encapsulated MIN6 cells within the DN HA/alginate hydrogel up to 3 days. Results are shown as mean \pm standard deviation (S.D.) with sample size $n=5$. (g) Representative photographs of the scaffolds made of DN HA/alginate hydrogel (i) before the transplantation, (ii) after being retrieved at 30 days. Scale bar, 1 mm.

Universality of DN (A+B)/C hydrogels as bio-inks for DASP 2.0

We note that the cytocompatibility of the PAM/alginate DN bio-ink is relatively low, with cell viability ~60% (**Fig. 5c**), largely because free PAM-based polymers are not completely cytocompatible (**Supplementary Fig. 15**). Yet, the concept of (A+B)/C DN hydrogels based on TZ-NB click chemistry is expected to be applicable to other biopolymers, where A, B, and C respectively represent TZ-polymer, NB-polymer, and alginate. To test this, we replace the PAM polymer by hyaluronic acid (HA), a natural polysaccharide widely used in biomaterials development. Following the same chemistry as for the PAM (**Fig. 1b**), we functionalize a pair of HA polymers, respectively, with TZ and NB at nearly the same grafting ratio of 27%, corresponding to approximately 250 TZ or NB groups per polymer (**Supplementary Figs. 6, 7**). Upon mixing, the pair of TZ-HA and NB-HA crosslink to form a network via additive-free click reaction, as illustrated in **Fig. 6a**.

Compared to the single-network hydrogels, the improvement in the mechanical properties of the HA/alginate DN hydrogel is comparable to that observed for the PAM/alginate formulation. For instance, the shear modulus of a single-network HA is about 500 Pa (solid grey line, **Fig. 6b**). By contrast, for the DN HA/alginate hydrogel, G is about 3050 Pa, approximately the sum of HA and alginate single-network hydrogels (orange solid line, **Fig. 6b**). Moreover, unlike the single-network HA hydrogel that is brittle with a tensile breaking strain of 0.6, the DN HA/alginate hydrogel is significantly more stretchable with $\epsilon_f = 1.02$, as shown in **Figs. 6c, d** and **Supplementary Video 7**.

As expected, the HA/alginate is suitable for DASP printing, as demonstrated by the success of creating a 5×5×4 lattice scaffold (**Fig. 6e**). To test the cytocompatibility of the HA/alginate hydrogel, we directly mix MIN6 cells with the bio-ink at a cell density of 5 million/mL. Live/dead assay reveals that the viability of HA/alginate encapsulated MIN6 cells remains above 80% for three days after the printing (**Fig. 6f**). This survival rate is significantly higher than 60% observed in PAM₁₀Alg₂, and importantly, meets the standards for most applications¹. Finally, we transplant the DASP printed 5×5×4 lattice scaffolds made of HA/alginate into the mouse abdominal cavity. The integrity of the retrieved scaffolds is confirmed 30 days after transplantation (**Fig. 6g**), indicating that HA/alginate DN scaffolds are sufficiently robust to maintain long-term *in vivo* stability. Taken together, these results demonstrate that the concept of (A+B)/C hydrogels provides a universal strategy for the development of modular DN bio-inks for DASP printing.

In context of cell viability, our results show that the cell viability is significantly improved without sacrificing mechanical robustness by replacing the polymer backbone. In context of encapsulation approach, these new results show that the cells can be directly premixed with the bio-inks without significantly impairing cell viability. Further, in the HA/Alg formulation, we purposely design the A+B bio-inks with a mismatched NB/TZ ratio of 2/1, resulting in considerable unreacted NB groups. In the future, we will exploit these unreacted NB groups to conjugate ECM derived peptides for cell engagement and degradable crosslinkers for 3D cell culture.

11. Regarding viability of cells: how does the surface compare to the core of droplets? As the method relies on the encapsulation within a more dense bulk material (with microporosity), viability might be dependent on the diffusion distance. A characterization of cell viability vs. localization within droplets (and differences in viability/metabolic activity) is important. The authors should also comment on physical limitations (diffusion limits, nutrient availability) in single droplets vs. aggregate structures. Herein, it would be helpful to quantify the viability limit from the outside of the droplets.

Our previous studies² based on 4% pure alginate with comparable stiffness to that of PAM₁₀Alg₂ show that the size of droplets is small enough for efficient nutrient transport, and no significant difference in cell viability along the droplet diameter was observed (**Fig. R2**).

[REDACTED]

Fig. R2. A representative fluorescence confocal microscopy image from live/dead assay of DASP encapsulated MIN6 cells. The dashed lines outline the interparticle space. Adapted from Ref. ²

12. Line 528 should read: “more cytocompatible” as there is still relatively low viability exhibited in the crosslinked DN.

We have replaced the original statement with “Although this caveat can be mitigated using the triplet-inlet print nozzle to avoid directly mixing the cells with the PAM polymers for a long time, the cytocompatibility of the bio-ink remains relatively low with a cell viability of 60%.” in the main text.

13. The authors state that the scaffold can be retrieved from the implantation site of immunocompetent mice after 4 months without significant impairments to the structural integrity. A photograph of the construct before and after is shown. However, the reviewer is missing any kind of histological assessment showing (immune) cell infiltration, blood vessel infiltration, matrix deposition etc. As no significant changes in mechanical characteristics (stress-strain) were observed following the *in vivo* explantation (Fig 5h), the scaffold materials seem rather inert to *in vivo* environments.

We appreciate the reviewer’s comments. This work focuses on the development of DN bio-inks for DASP printing of mechanical robust scaffolds. To this end, the goal of the *in vivo* study is solely to demonstrate *in vivo* stability of DASP printed DN scaffolds. The interaction between cells and biomaterials such as cell infiltration, blood vessel infiltration, and matrix deposition are great topics for future explorations but are beyond the scope of the current work.

Minor comments:

- Fig 3e: There are two “PAM10Alg2”

The empty circle is Empty circle corresponds to the hydrogel PAM₁₀Alg₂ with mismatched TZ and NB grafting ratios, as described the figure caption in original submission. We have made the legend clearer in the revised figure.

- Line 121: “[...] decreases with rapidly with [...]” should be “decreases rapidly with”

- Line 263: 2.5% TZ and 0.5% NB should give you NB/TZ=0.2, instead of 0.2 in Fig. 2a

We thank reviewer for catching this typo. The NB grating ratio is actually 0.63% tested by NMR. Therefore, the NB/TZ should be 0.25 rather than 0.2. We have replaced the “0.5%” to “0.63%” and replaced the “PAM₁₀^{1/5}Alg₂” to “PAM₁₀^{1/4}Alg₂”.

- Line 419: Release index = low/high or high/low glucose media?
- Line 532: “[...] relies on-demand [...]”]; missing “on” after relies
- Line 539: “[...] enabling the fabrication highly complex [...]”]; missing “of” after fabrication

We appreciate the reviewer for catching these and we have corrected all of them in the revised manuscript.

Reviewer #2

The authors have introduced the novel concept of Digital Assembly of Spherical Particles (DASP) as a means of generating larger volumes of engineered tissue constructs through the utilization of voxelated bioprinting technology. This particular methodology has undergone comprehensive development and establishment by the authors, culminating in the publication of multiple articles in various academic journals. The principal innovation of this study lies in the application of a new ink material aimed at yielding more flexible structures as opposed to the single-network Polyacrylamide (PAM) hydrogel.

The objective is to match the mechanical properties of the engineered tissue constructs with those of native tissues and organs. Nevertheless, a clear rationale for the necessity of elasticity in the bioink material remains elusive.

We thank the reviewer for the complementary remarks about our work. We would like to clarify that the goal of this study is not solely matching the mechanical properties of the engineered tissue constructs with those of native tissues and organs. Instead, the major goal is to further advance voxelated bioprinting through a combination of biomaterials and technology innovation. To this end, we have developed a substantially improved voxelated bioprinting technology (DASP 2.0) that enables the digital assembly of interpenetrating double-network bio-ink droplets.

The developed biomaterials, technology, and systems have far-reaching implications in biomanufacturing. In the context of biomaterials, the concept of using a pair of stoichiometrically matched functionalized polymers for hydrogel synthesis offers precise control over crosslinking kinetics and stiffness. Moreover, the additive-free click chemistry allows the bio-inks to be solidified before removing the supporting matrix, enabling the fabrication highly complex 3D objects with high fidelity. In the context of hardware engineering, the capability of integrating multiple flows into a single print nozzle offers versatility in processing bio-inks, so that the bio-inks can be printed and crosslinked *in situ* without external interventions. This capability may be appealing for administering biomaterials *in vivo*, which is an environment with limited access to external triggers. Finally, the developed multi-channel print nozzles can be easily adapted to medical syringes and needles for administering the DN bio-inks as a new class of injectable biomaterials.

The motivation for developing the DN bio-ink derives from the limited mechanical properties of the pure alginate bio-ink we used in the DASP 1.0. In previous studies, the pure alginate hydrogels are brittle and have limited tunability in extensibility, which constrain the applications of DASP not only in translational biomedicine, such as printing mechanically robust scaffolds as long-term transplants, but also in basic biomedical research, which requires modular biomaterials for controlling the microenvironment of voxels. By contrast, the DN hydrogel not only provides DASP printed structures with sufficient robustness to withstand external forces, but also enables modular control over the stiffness and extensibility.

We appreciate the reviewer that the study should be better motivated to enhance the overall clarity and impact. In the **Introduction** of revise manuscript, we have added comprehensive discussions on the difference between DASP and other 3D printing technologies for manipulating spherical objects, as detailed in the reply to **Comment #2** from **Reviewer #3**. In addition, we added a sentence to better motivate our work:

“However, our prototype voxelated bioprinting technology can print pure alginate hydrogels only, which are brittle and have limited tunability in mechanical properties. These limit the applications of DASP not only in translational biomedicine, such as printing mechanically robust scaffolds as

long-term transplants, but also in basic biomedical research, which requires modular biomaterials for controlling the microenvironment of voxels.”

The authors asserted that the DASP hydrogel is created by crosslinking PAM polymers via a biologically compatible, bioorthogonal chemistry. However, it is noteworthy that the DASP hydrogel does not directly encapsulate cells, which raises questions regarding the unequivocal characterization of its "biocompatibility."

This is an excellent comment. Similar comment was also raised by **Reviewer #1** in **Comment #10**. Please see above for detailed reply.

Furthermore, the definition of bioorthogonal chemistry implies a direct interaction between specific cellular sites and the material, a criterion not met by the DASP hydrogel, potentially constituting an overstatement.

That is an excellent suggestion. In the initial submission, we intended to use “bioorthogonal” to describe a cytocompatibility of TZ-NB click chemistry, considering a lot of click chemistry requires biohazard UV or copper ions for triggering. In revised version, we have replaced all “bioorthogonal chemistry” to “biofriendly click chemistry”.

Despite the authors' comprehensive elucidation of the DASP hydrogel within the context of 3D printing, its attention from the bioprinting research field is hindered by its limited viability. Even with the addition of a cell-laden layer, the system failed to meet the expected viability standards post-printing. Additionally, the inherent properties of the applied materials precluded the anticipation of significant biological functionality.

We thank the reviewer for raising these excellent comments. To address these, in the revised manuscript, we have developed a new HA/Alg bio-ink with significantly improved viability, as detailed in the reply to **Comment #10** by **Reviewer #1**.

For potential biological functionality, during the development of the HA/Alg bio-ink, we intentionally mix TZ-HA and NB-HA with a mismatched NB/TZ ratio of 2/1 (**Fig. 6a**), resulting in considerable unreacted free NB groups in the hydrogel network. These free NB groups can be used to conjugate extracellular matrix derived peptides for cell engagement. Moreover, degradable crosslinkers and growth factors may be conjugated into the hydrogel network to promote cell migration and vascularization. The associated discussion has been added to the **Discussion and outlook** section of the revised manuscript.

“We note a few limitations and potential challenges associated with applying DASP 2.0 for basic and translational biomedicine. For instance, to exploit DASP 2.0 to understand and control cell-matrix interactions, it is necessary to render the DN hydrogels cell-instructive. To this end, when preparing the DN HA/alginate hydrogel, the TZ-HA and NB-HA are mixed with a mismatched NB/TZ ratio of 2/1 (**Fig. 6a**), resulting in considerable unreacted free NB groups in the hydrogel network. These free NB groups may be used to conjugate extracellular matrix derived peptides for cell engagement ^{4,5}. Moreover, degradable crosslinkers ^{6,7} and growth factors ⁸ may be conjugated into the hydrogel network to promote cell migration and vascularization.”

In light of these considerations, this manuscript must undergo substantial refinement to enhance its originality and to furnish concrete scientific findings pertaining to the bioink's suitability for utilization in bioprinting before its consideration for publication.

We thank the reviewer again for constructive comments. We hope that our new experiments have addressed these concerns.

Reviewer #3

1. This manuscript discusses progress in the development of biomaterial ink, the technology used for printing it, and the possible applications of the material in the form of 3D objects. The author highlights advancements in printing voxels based on alginate and PAM, representing an improvement over the previously developed DASP1, which could only print alginate. However, DASP1 faced toxicity issues due to unreacted polyacrylamide. The authors also present in vivo data assessing the safety of the 3D printed materials. The article provide interesting information and add to the body of the literature but shall require major revision before reconsideration.

2. The authors are encouraged to provide a more thorough discussion of their previous development, DASP1, as its significance remains unclear to the reviewer. The study's importance should extend beyond the comparison of PAM/alginate versus alginate and the hardware development for printing the mixture. A more comprehensive discussion in this regard would enhance the overall clarity and impact of the study.

We thank the reviewer for this excellent suggestion. In the **Introduction** of the revised manuscript, we have highlighted the difference between DASP and existing 3D printing techniques for manipulating spherical objects to enhance the clarity and impact of the study.

DASP generates a highly viscoelastic aqueous droplet in an aqueous yield-stress fluid, deposits the droplet at a prescribed location, and assembles individual droplets into 3D structures by controlled polymer swelling (Fig. 1a). DASP is different from existing 3D printing techniques for manipulating spherical objects such as droplets and spheroids. For example, in embedded droplet printing, individual droplets are dispensed far apart from each other to form discrete patterns⁹⁻¹² or randomly stacked together to form thick structures without control over particle location and inter-particle distance¹³. By contrast, DASP enables on-demand generation, deposition, and assembly of individual bio-ink droplets to form integrated, organized 3D structures. Moreover, DASP enables printing highly viscoelastic, non-Newtonian bio-inks in a cytocompatible environment. This is in a stark contrast to embedded droplet printing that relies on the interfacial tension between the immiscible aqueous and oil-like fluids to generate droplets, which, therefore, can handle low viscosity fluids only^{9,10}. Compared to the bioassembly of cell aggregates or spheroids¹⁴⁻¹⁶ where cells are often subject to direct mechanical manipulation, in DASP cells are encapsulated in a hydrogel matrix, which avoids possible high mechanical shear-induced cell damage. Furthermore, the bio-ink particles can be assembled in a prescribed manner in a short time through controlled polymer swelling. This contrasts with the sophisticated surface chemistry such as lipid bilayers required to join droplets^{17,18} and the slow cell migration required to fuse spheroids^{15,16}. Thus, DASP provides a facile, robust approach for the on-demand generation, deposition, and assembly of highly viscoelastic hydrogel particles in a cytocompatible environment. Using DASP, we created multiscale porous scaffolds formed by interconnected yet distinguishable alginate hydrogel particles. In our proof-of-concept research, we encapsulated human islets in individual particles and demonstrated that the scaffolds allow for responsive insulin release², highlighting the potential of DASP in biomedical applications.

3. The reported method appears primarily suited for generating small size constructs. Has the author explored the size variability associated with this method? It would be valuable to investigate the range of sizes feasible through this approach, specifically identifying the maximum and minimum constructs that can be 3D printed and shall be compared with their DASP1. Additionally, it is essential to elucidate the material dependency of this approach concerning the dimensions of the constructs.

We agree with the reviewer that all these questions are valuable to explore. But they are beyond the scope of this paper. We are actively researching in this area and hope to present our findings to the community in the future.

Nevertheless, we appreciate the reviewer's comment on the size of the 3D printed construct. The size of voxelated 3D constructs is determined by (1) the size of individual voxels (droplets) and (2) the number of voxels that can be printed. As of now, the lower size limit of a voxel is about 300 μm in diameter, as shown in our previous work². Based on our experience, it is possible to reduce the droplet size to ~ 100 μm by using glass capillary nozzles with a small nozzle diameter, but difficult to print droplets < 50 μm with good fidelity because of the difficulty associated with detaching the nozzle from a small highly viscoelastic droplet³.

In principle, the number of voxels is not limited by the chemistry but by the size of the supporting matrix. However, the stability of the 3D printed structure depends on the mechanical properties of the bio-ink. Compared with pure alginate, although the new chemistry of the DN bio-inks does not enable printing smaller structures nor better printing resolution, it enables creating more complex structure due to the improved mechanical robustness of the DN hydrogel, as demonstrated by the printed hollow sphere in the original submission.

4. The details regarding the mechanical properties of the constructs, particularly in terms of compression, need to be more thoroughly elaborated with respect to the size of the scaffolds. It is crucial to investigate whether the voxel crosslinking is linearly influenced by the size of the scaffold, potentially leading to variations in compression modulus. Additionally, exploring how compression force may impact the interaction of voxels becomes pertinent. A more in depth analysis of these aspects may improve the understanding of the interplay between scaffold size, voxel crosslinking, and compression behavior, providing valuable insights into the mechanical characteristics of the constructs.

We appreciate the reviewer's comments on comprehensive study of the mechanical properties of the constructs. These are important topics in the context of mechanics of voxelated scaffolds, which are different from the bulk counterparts in that the individual voxels are interconnected yet remain distinguishable. However, systemic exploration of the mechanics of voxelated constructs is beyond the scope of this work, which focuses advancing voxelated bioprinting technology through a combination of biomaterials and technology development. We hope to present these studies to the community in the future.

5. While the printing of voxels introduces multiscale porosity, potentially facilitating nutrient diffusion into the scaffold, I couldn't see the data related to construct porosity. It would be valuable to include information on the porosity of the construct, particularly examining both macro and micro porosity. Utilizing uCT and SEM may be helpful in this regard. This additional information is crucial for a comprehensive understanding of the scaffold's porosity characteristics, and its suitability for nutrient diffusion and overall performance.

The macroscopic porosity, or inter-particle space, of DASP printed structure was demonstrated in our previous work using digital camera and confocal microscopy (**Fig. R1**). For the microscopic porosity, we have added experiments to further characterize the porosity of the two-phase system in 3D, the details of which can be found in the reply to **Comment #9** by **Reviewer #1**.

6. The animal study is insufficient to establish the material's usefulness. While it demonstrates the lack of toxicity, the significance of this approach should extend beyond a mere safety test. Notably, there is an absence of data on whether the porosity contributes to vascularization within the construct? To fully show the application and performance of the reported material/method in vivo, it is crucial to include

comprehensive data addressing aspects such as vascularization. This will strengthen the study's overall impact.

This reviewer raised an excellent point – whether the multiscale porosity of voxelated 3D scaffolds contributes to the vascularization within the construct? Indeed, that is exactly the question we plan to investigate as the next step but is beyond the scope of this work. The goal of this current work is to demonstrate a significantly advanced voxelated bioprinting technology (DASP 2.0), and that the animal study is designed for only demonstrating the mechanical robustness of the printed DN scaffolds. We are actively exploiting voxelated bioprinting technology for both basic and translational biomedicine, including understanding the roles of multiscale porosity in vasculature, and hope to share our results with the community soon.

7. In line with previous comment what was/is the fate of the material that implanted in vivo? It was mentioned that the implant was retrieved after 7 days, first the 7 day window may be too short for the analysis. Second I don't see the histology data, or any immunohistochemistry information. Do we know if the porous structure remained permeable during the study? [continued]

We would like to clarify that the 7-day period is only for scaffolds made of pure alginate, as we find that the scaffolds have already been disassociated and almost disappeared by then. By contrast, the scaffolds made of DN hydrogel maintain their structural integrity up to 120 days, a period sufficient for testing long-term material stability, as shown in the original submission. Because the constructs remain nearly intact in structure, they remain multiscale porous and permeable.

Since the focus of this study is to develop a new kind of mechanically robust bio-ink for voxelated bioprinting, we would like to save the histology study for future explorations, in which we plan to systematically explore the biocompatibility and function of cell encapsulated scaffolds.

What are the swelling properties of the voxels? There are many data missing both for in vivo studies and also concerning the fate of the material in vitro (swelling, degradation plus the mechanical studies of the material behaviour in vitro e.g. in SBF) and in vivo. If degraded then the rate and kinetic of degradation need to be provided.

The crosslinked PAM₁₀Alg₂ DN hydrogel swells by 7% in diameter after being incubated in DMEM for 24 h, as shown in **Supplementary Fig. 8** and described in the revised manuscript.

Supplementary Fig. 8. Swelling of a bulk DN hydrogel. Photos of the PAM₁₀Alg₂ DN hydrogel in DMEM media up to 24h. Scale bar, 10 mm.

“we equilibrate and completely crosslink hydrogel in cell culture media (DMEM) with 2 mM CaCl₂ for 24 hours before characterizing its mechanical properties. Indeed, all the hydrogel samples

swell in DMEM, as exemplified by a swelling ratio of 7% in size for PAM₁₀Alg₂ (Supplementary Fig. 8). Consequently, the shear modulus of PAM₁₀ hydrogel decreases...”

In this study, we are using permanent covalent crosslinks formed by the click reaction between tetrazine and norbornene groups to solidify the hydrogel. Thus, the scaffold is not degradable, as evidenced by the nearly the same strain-stress curves at different time points after transplantation (Fig. 5h).

8. The rationale behind employing on-demand mixing of the TZ-PAM and NB-PAM polymers for printing the DN bio-300 ink is not clearly defined. The authors should first show the necessity of this specific design and subsequently address its applicability to other bioinks. It is crucial to articulate the importance of undertaking such actions in comparison to their previously developed alginate-only method. doing so may enhance the reader's understanding of the motivation behind the chosen approach and its potential advantages over alternative methods, particularly in relation to their prior work with alginate-only materials.

We appreciate the reviewer for the suggestion and have added a paragraph to clarify the motivation for on-demand mixing of the A+B polymers.

“For DASP printing, the bio-inks are required to be extruded through the nozzle in a liquid state and rapidly crosslink once being deposited. Due to the rapid crosslinking of DN bio-inks upon mixing (Fig. 2a), premixing and loading the DN bio-inks into a single syringe, as performed in DASP 1.0, would clog the print nozzle and syringe. To avoid the clogging, we separately load the TZ-PAM and the NB-PAM in two syringes and then mix them on-demand during the printing. To do so, we engineer a dual-inlet print nozzle with an inner static mixer chamber ”

9. In the introduction section, the manuscript did not provide a clear and comprehensive literature review on the current state-of-the-art. While the importance of voxel printing was compared with droplet printing and the importance of printing viscoelastic ink in this approach but the authors could also indicate the microgel printing approach or printing the emulsion based biomaterial ink that also can result in constructs with good mechanical properties and may require a simpler approach compared to printing voxels. In general providing advantages and disadvantages of different biomaterials and printing methods, and the gaps and challenges that the proposed biomaterial ink aims to overcome.

We thank the reviewer for raising this excellent point. To address the reviewer's comment, in the revised manuscript, we have added a comprehensive discussion on the difference between DASP and existing technologies for manipulating spherical objects. More details can be found in the reply to **Comment #2** from **Reviewer #3**.

10. The authors encapsulated human islets in individual particles and demonstrated that the scaffolds allow for responsive insulin release to highlight the potential of DASP in biomedical applications, nevertheless there is a wide range of studies on encapsulating human islets in gel beads including alginate based materials developed by simple bead formation., microfluidic beads and so on with a diverse range of control on the porosity of the beads to induce release of insulin while preventing the damage to the encapsulated islets. While there is no information here on the nutrient and oxygen diffusion to the scaffolds not the role of the scaffold in protecting the cells from immune system, nor also on cellular overgrowth, then the authors need to carefully revise, adding limitation to their studies and provide a better discussion compared to the existing state of the art. Then the authors need to better discuss the potential application of the DASP realistically.

We would like to clarify that human islets were used in our previous work². In this study, we use MIN6 cells rather than islets for *in vitro* tests. As the MIN6 cells retains the function of glucose-

stimulated insulin secretion, we use them to test the cytocompatibility and transport properties of the DASP scaffolds, as described in the original submission.

Additionally, as this reviewer pointed out, DASP has the potential of being applied to encapsulate islets to treat type 1 diabetes. This is a direction that we have been actively pursuing but is beyond the scope of this work. Nevertheless, we appreciate the reviewer's comment and have added discussions on the potential and limitations of DASP technology in cell-based therapy in the **Conclusion and Outlook** section. Please see the reply to **Comment #12** for details.

11. The authors also should better define and justify the ratio of TZ-PAM and the NB-PAM and alginate and its connection to the printability of the biomaterial ink. Also the crosslinking density of the material is not clear. Has there been an optimisation study?

DASP printing requires highly viscous and shear-thinning bio-inks, such that during printing, an extruded droplet can grow uniformly in the yield-stress fluid supporting matrix³ (line 286 in the original submission). Moreover, the bio-ink should possess a gelation time of sufficient duration. This allows the printed droplets to undergo partial swelling, enabling them to partially coalesce with adjacent droplets before complete crosslinking occurs (**Fig. 1a**). However, the crosslinking time should not be too long, otherwise the bio-ink may diffuse through the porous supporting matrix, leading to uncontrollable droplet shape² (lines 78-82 in the original submission).

We use PAM₁₀Alg₂ as the DN bio-ink for DASP printing because this formulation has a desirable gelation time of 40 sec and rheological properties. Moreover, the crosslinked DN hydrogel has a stiffness comparable to that of the organs in abdominal cavity and a significantly improved extensibility compared to its single-network constituents.

To address the reviewer's comments, in the revised manuscript we have further clarified the rationale of the DN bio-ink formulation for DASP printing.

“These properties make the DN bio-ink PAM₁₀Alg₂ suitable for DASP printing³. Moreover, the crosslinked DN hydrogel has a significantly improved extensibility compared to its single-network constituents and a stiffness comparable to that of the organs in the abnormal cavity. Thus, we use PAM₁₀Alg₂ as the DN bio-ink for the subsequent DASP printing.”

12. In the conclusion section, the authors should also acknowledge or address some potential issues or challenges of the study, such as the reproducibility and scalability of the printing method, the biocompatibility and immunogenicity of the biomaterial ink and, the long-term stability and functionality of the constructs, and the clinical relevance and applicability of the biomaterial ink for biomedical engineering applications. Therefore, I recommend that the paper be revised to address these limitations and challenges before it can be accepted for publication.

We thank the reviewer for this excellent suggestion. In the revised manuscript, in the **Conclusion and outlook** section, we have added two paragraphs to discuss the limitations and potential challenges associated with applying DASP 2.0 for basic and translational biomedicine.

“We note a few limitations and potential challenges associated with applying DASP 2.0 for basic and translational biomedicine. For instance, to exploit DASP 2.0 to understand and control cell-matrix interactions, it is necessary to render the DN hydrogels cell-instructive. To this end, when preparing the DN HA/alginate hydrogel, the TZ-HA and NB-HA are mixed with a mismatched NB/TZ ratio of 2/1 (**Fig. 6a**), resulting in considerable unreacted free NB groups in the hydrogel network. These free NB groups may be used to conjugate extracellular matrix derived peptides

for cell engagement ^{4,5}. Moreover, degradable crosslinkers ^{6,7} and growth factors ⁸ may be conjugated into the hydrogel network to promote cell migration and vascularization.

In the context of translational biomedicine, our results highlight the potential of voxelated bioprinting in manufacturing cell-encapsulated multiscale porous and mechanically robust scaffolds as therapeutic transplants ^{19,20}. For instance, a potential application is to encapsulate human islets in a voxelated 3D scaffold, which may be used as a transplant to reverse type 1 diabetes. Since a voxelated 3D scaffold consists of interconnected yet distinguishable hydrogel particles, it may combine the advantages of classical microencapsulation technology in high surface-to-volume ratio ^{21,22} and macro-devices in easy retrievability ^{23,24}. However, it remains to be determined the biomaterial biocompatibility and the ability of voxelated 3D scaffolds to allow long-term cell survival and function. Nevertheless, DASP 2.0 represents a significantly advanced voxelated bioprinting technology, paving the way for engineering highly complex yet organized functional 3D tissue constructs.”

References

1. Murphy, S. V & Atala, A. 3D Bioprinting of Tissues and Organs. *Nat. Biotechnol.* **32**, 773–785 (2014).
2. Zhu, J. *et al.* Digital Assembly of Spherical Viscoelastic Bio-Ink Particles. *Adv. Funct. Mater.* **32**, 2109004 (2022).
3. Zhu, J. & Cai, L. H. All-aqueous printing of viscoelastic droplets in yield-stress fluids. *Acta Biomater.* **165**, 60–71 (2023).
4. Frith, J. E., Mills, R. J., Hudson, J. E. & Cooper-White, J. J. Tailored integrin-extracellular matrix interactions to direct human mesenchymal stem cell differentiation. *Stem Cells Dev.* **21**, 2442–2456 (2012).
5. Caliarì, S. R., Vega, S. L., Kwon, M., Soulas, E. M. & Burdick, J. A. Dimensionality and spreading influence MSC YAP/TAZ signaling in hydrogel environments. *Biomaterials* **103**, 314–323 (2016).
6. Qazi, T. H. *et al.* Programming hydrogels to probe spatiotemporal cell biology. *Cell Stem Cell* **29**, 678–691 (2022).
7. Raeber, G. P., Lutolf, M. P. & Hubbell, J. A. Molecularly engineered PEG hydrogels: A novel model system for proteolytically mediated cell migration. *Biophys. J.* **89**, 1374–1388 (2005).
8. Ekaputra, A. K., Prestwich, G. D., Cool, S. M. & Hutmacher, D. W. The three-dimensional vascularization of growth factor-releasing hybrid scaffold of poly (ε-caprolactone)/collagen fibers and hyaluronic acid hydrogel. *Biomaterials* **32**, 8108–8117 (2011).
9. Cai, L., Marthelot, J. & Brun, P. T. An unbounded approach to microfluidics using the Rayleigh–Plateau instability of viscous threads directly drawn in a bath. *Proc. Natl. Acad. Sci. U. S. A.* **116**, 22966–22971 (2019).
10. Nelson, A. Z., Kundukad, B., Wong, W. K., Khan, S. A. & Doyle, P. S. Embedded droplet printing in yield-stress fluids. *Proc. Natl. Acad. Sci. U. S. A.* **117**, 5671–5679 (2020).
11. Foresti, D. *et al.* Acoustophoretic printing. *Sci. Adv.* **4**, 1–10 (2018).
12. Zhang, Z., Xiong, R., Corr, D. T. & Huang, Y. Study of Impingement Types and Printing Quality during Laser Printing of Viscoelastic Alginate Solutions. *Langmuir* **32**, 3004–3014 (2016).
13. Mea, H. J., Delgadillo, L. & Wan, J. On-demand modulation of 3D-printed elastomers using programmable droplet inclusions. *Proc. Natl. Acad. Sci. U. S. A.* **117**, 14790–14797 (2020).
14. Mironov, V. *et al.* Organ printing: Tissue spheroids as building blocks. *Biomaterials* **30**, 2164–2174 (2009).
15. Ayan, B. *et al.* Aspiration-assisted bioprinting for precise positioning of biologics. *Sci. Adv.* **6**, 1–17 (2020).
16. Daly, A. C., Davidson, M. D. & Burdick, J. A. 3D bioprinting of high cell-density heterogeneous tissue models through spheroid fusion within self-healing hydrogels. *Nat. Commun.* **12**, 1–13 (2021).
17. Downs, F. G. *et al.* Multi-responsive hydrogel structures from patterned droplet networks. *Nat. Chem.* **12**, 363–371 (2020).
18. Villar, G., Graham, A. D. & Bayley, H. A tissue-like printed material. *Science (80-.)*. **340**, 48–53 (2013).
19. Gupta, P., Alheib, O. & Shin, J. Towards single cell encapsulation for precision biology and medicine. *Adv. Drug Deliv. Rev.* 115010 (2023) doi:10.1016/j.addr.2023.115010.
20. Bashor, C. J., Hilton, I. B., Bandukwala, H., Smith, D. M. & Veisheh, O. Engineering the next generation of cell-based therapeutics. *Nat. Rev. Drug Discov.* **21**, 655–675 (2022).
21. Bochenek, M. A. *et al.* Alginate encapsulation as long-term immune protection of allogeneic pancreatic islet cells transplanted into the omental bursa of macaques. *Nat.*

- Biomed. Eng.* **2**, 810–821 (2018).
22. Ernst, A. U., Wang, L. H. & Ma, M. Islet encapsulation. *J. Mater. Chem. B* **6**, 6705–6722 (2018).
 23. Song, S. & Roy, S. Progress and challenges in macroencapsulation approaches for type 1 diabetes (T1D) treatment: Cells, biomaterials, and devices. *Biotechnol. Bioeng.* **113**, 1381–1402 (2016).
 24. Lee, S. H. *et al.* Human β -cell precursors mature into functional insulin-producing cells in an immunoisolation device: Implications for diabetes cell therapies. *Transplantation* **87**, 983–991 (2009).

REVIEWERS' COMMENTS

Reviewer #1 (Remarks to the Author):

The authors have substantially revised the manuscript in response to reviewer comments. Of note is the new Fig. 6 demonstrating a ~80% viability using an alginate/click-HA system, a significant improvement over the prior ~60% viability.

While I still feel that further relatively straight-forward in vivo studies to assess immune/foreign body response and host cell invasion/migration would significantly strengthen the case for these materials and biofabrication approach, it is clear that the manuscript in its present form already presents substantial new advances and that these studies may best form part of a future manuscript.

I congratulate the authors for a thorough revision, and recommend the manuscript for publication in its current form.

Reviewer #3 (Remarks to the Author):

[Editorial Note: Reviewer #3 was also asked to look over the responses given to Reviewer #2]

The authors sufficiently addressed all comments, including those raised by the second reviewer regarding the cytocompatibility of the materials and the rationale of the study. In the revised manuscript, the authors revised the introduction and also conducted additional experiments, and developed a new set of TZ and NB functionalized polymers by substituting the PAM backbone with HA. This modification led to enhanced cell viability without compromising mechanical strength. In my view, the paper is now suitable for publication.